# LEARNING THE QUADRATIC ASSIGNMENT PROBLEM WITH WARM-STARTED MCMC FINETUNING AND CROSS-GRAPH ATTENTION

## ABSTRACT

The quadratic assignment problem (QAP) is a fundamental NP-hard task that poses significant challenges for both traditional heuristics and modern learning-based solvers. Current neural solvers for the QAP often struggle with poor scalability or limited flexibility when dealing with the coupled two-graph structure, and they underperform on real-world instances. We propose PLMA, an innovative permutation learning framework to bridge this performance gap. PLMA features an efficient warm-started MCMC finetuning procedure to enhance deployment-time performance, leveraging short Markov chains to anchor the adaptation to the promising regions previously explored. For rapid exploration via MCMC, we design an additive energy-based model (EBM) over the permutation space, which enables an $O(1)$-time 2-swap Metropolis-Hastings sampling step. Moreover, the network used to parameterize the EBM incorporates a cross-graph attention mechanism that directly models the coupled graph structure of the QAP, ensuring scalability and flexibility. Extensive experiments demonstrate the consistent superiority of PLMA over stat-of-the-art baseline methods across various benchmarks, highlighted by a near-zero average optimality gap on QAPLIB and remarkably superior robustness on the notoriously difficult Taixxeyy instances.

## 1 INTRODUCTION

The quadratic assignment problem (QAP) is one of the most fundamental combinatorial optimization (CO) problems with a wide range of real-world applications, such as facility location (Elshafei, 1977), keyboard design (Burkard, 1984) and graph matching (Caetano et al., 2009). The problem seeks to assign $n$ facilities to $n$ locations to minimize the total cost, defined by the sum of products of flows between facilities and distances between their assigned locations. Despite of its broad applications, the QAP is not only NP-hard to solve but also NP-hard to approximate (Sahni & Gonzalez, 1976), rendering exact optimality practically intractable. Exact solvers like branch-and-bound and cutting plane methods struggle to solve instances of size larger than $n = 20$ within a reasonable time budget (Li et al., 1994). Handcrafted heuristics such as tabu search (Taillard, 1995) and simulated annealing (Burkard & Rendl, 1984) can find high-quality suboptimal solutions, but they require extensive instance-specific tuning and generalize poorly across distributions.

Machine learning offers a potential avenue to learn problem structure and reuse knowledge across instances. In recent years, learning-based methods have achieved notable success on CO problems such as routing (Kool et al., 2018; Kwon et al., 2020; Wu et al., 2021; Zhou et al., 2024; Li et al., 2025) and scheduling problems (Zhang et al., 2020; Wang et al., 2021a; Song et al., 2022; Jeon et al., 2023). Much of this success leverages single-graph representations that align naturally with expressive neural architectures. However, the unique structure of the QAP presents distinct challenges. Specifically, a QAP instance is defined by two interdependent matrices, a flow matrix and a distance matrix, which can be viewed as two distinct graphs. Solving the QAP thus necessitates a sophisticated mechanism to fuse information across these two graphs. Existing neural methods typically approach this by either constructing an association graph with $n^2$ nodes (Wang et al., 2021b; Liu et al., 2023) or by concatenates facility and location embeddings via a complete permutation (Tan & Mu, 2024). However, the former suffers from poor scalability, and the latter lacks generality as it presupposes the existence of a full solution. Overall, the performance of learning-based methods on

real-world QAP instances remains far from satisfactory, with average optimality gaps up to $37.82\%$ on QAPLIB (Burkard et al., 1997).

To bridge this critical performance gap, we propose **PLMA**, a **P**ermutation **L**earning framework featuring a warm-started **M**arkov chain Monte Carlo (MCMC) finetuning procedure along with a cross-graph **A**ttention mechanism. Our approach establishes a new state-of-the-art for learning-based QAP solvers. Our main contributions are summarized as follows. (1) We propose a two-stage learning paradigm that first pretrains a model on diverse instances to learn transferable structural priors, and then adapts it to target instances with a highly efficient, batch-wise MCMC finetuning procedure. Within this learning paradigm, the finetuning procedure employs a unique warm-start mechanism that reuses previous high-quality solutions to initialize customized short and locally-interacted Markov chains, thereby focusing the adaptation on promising regions. (2) We design an energy-based model (EBM) whose additive structure facilitates remarkably efficient permutation space exploration. This design is tailored for MCMC, enabling $O(1)$-time evaluation of 2-swap proposals within a Metropolis-Hastings sampler. (3) We develop a scalable and flexible cross-graph attention mechanism to capture the coupled two-graph structure of the QAP. This mechanism directly models the interactions between the facility and location graphs, without the need of a large association graph or a full solution. (4) We demonstrate the state-of-the-art performance of PLMA through extensive experiments on various synthetic datasets and real-world benchmarks. Remarkably, PLMA not only achieves a near-zero average optimality gap on the canonical QAPLIB benchmark (Burkard et al., 1997), but also exhibits profound robustness on Taixxeyy instances (Drezner et al., 2005), a setting where leading heuristics are known to fail catastrophically.

## 1.1 RELATED WORK

**Learning-based methods for the QAP.** Learning-based methods for the QAP are relatively scarce in the literature. The majority of works focus on the application of QAP to graph matching (GM) (Nowak et al., 2018; Wang et al., 2019; 2021b; Liu et al., 2023). To capture the quadratic property, a popular approach is to combine the two input graphs into an association graph, whose vertices are node pairs and edge weights are affinity scores. NGM (Wang et al., 2021b) regards GM as a vertex classification task on the association graph and obtains the whole graph matching in one-shot. RGM (Liu et al., 2023) constructs a graph matching by sequentially selecting vertices from the association graph. The rely on the association graph, however, incurs significant computational cost. Towards directly solving the Koopmans-Beckmann's QAP and eliminating the need for an association graph, Tan & Mu (2024) propose a solution-aware Transformer, which learns swap local search operators to iteratively refine initial complete solutions.

**Deployment-time refinement.** A variety of works to improve deployment-time performance of neural CO solvers have emerged in recent years (Bello et al., 2016; Li et al., 2023; Xin et al., 2021; Grinsztajn et al., 2023; Chalumeau et al., 2023; Hottung et al., 2025). A central paradigm is active search (Bello et al., 2016), where model parameters are finetuned on test instances. Variants include EAS (Hottung et al., 2022), which updates only a subset of parameters or embeddings for efficiency, and DIMES (Qiu et al., 2022), which employs meta-learning to provide favorable initialization for finetuning. These active search methods are generally built upon auto-regressive models, which factorize the overall probability into a product of conditional probabilities and construct solutions sequentially in an element-wise manner. However, this process requires restarting from scratch at each refinement iteration, hindering effective reuse of previously identified high-quality solutions. In contrast, our work introduces a EBM with MCMC sampling, enabling warm-starts from prior solutions and overcoming the restart limitation of auto-regressive active search.

Although MCMC has been explored in MCPG (Chen et al., 2023), our method differs in two key aspects. First, MCPG is designed for binary optimization problems. A direct extension to QAP would require modeling the permutation matrix with binary variables and enforcing permutation constraints via penalty terms. This substantially enlarges the search domain beyond the feasible set, making the generation of valid assignments inefficient. In contrast, our method operates directly over the permutation space, ensuring that every sampled solution is inherently feasible. Second, the policy in MCPG is parameterized as an instance-agnostic multivariate Bernoulli distribution, hindering it from leveraging the transferable knowledge from pretraining. Our policy is parameterized by an instance-conditioned neural network, allowing the learning of general structural knowledge.

## 2 LEARNING TO SOLVE THE QAP

### 2.1 QUADRATIC ASSIGNMENT PROBLEM

In this paper, we consider the Koopmans-Beckann's QAP, which involves the optimal assignment of $n$ facilities to $n$ locations. A QAP instance $\mathcal{P}$ is specified by two $n \times n$ matrices with real elements $\boldsymbol{F} = (F_{ij}), \boldsymbol{D} = (D_{kl})$, where $F_{ij}$ is the flow between facilities $i$ and $j$ and $D_{kl}$ is the distance between locations $k$ and $l$. For any positive integer $n$, we denote the set $\{1, \ldots, n\}$ by $[n]$. Let $\Pi_n$ represent the set of all permutations over $[n]$. A permutation $\pi \in \Pi_n$ maps each facility $i$ to location $\pi(i)$. The QAP can then be formulated as

$$\min_{\pi \in \Pi_n} f(\pi; \mathcal{P}) := \sum_{i=1}^{n} \sum_{j=1}^{n} F_{ij} D_{\pi(i)\pi(j)}. \tag{1}$$

Alternatively, the QAP can be expressed equivalently in a matrix form

$$\min_{X \in \{0,1\}^{n \times n}} \langle F, XDX^{\mathrm{T}} \rangle, \quad \text{s.t.} \quad X\mathbf{1} = \mathbf{1}, X^{\mathrm{T}}\mathbf{1} = \mathbf{1}. \tag{2}$$

Here, the linear constraints restrict the binary matrix $X$ to the set of permutation matrices. For each $\pi \in \Pi_n$, its corresponding permutation matrix $X_\pi$ is defined by $(X_\pi)_{i,\pi(i)} = 1$ and zero otherwise. Then the equivalence to equation 1 is established by the identity $f(\pi, \mathcal{P}) = \langle F, X_\pi D X_\pi^{\mathrm{T}} \rangle$.

### 2.2 PROBABILISTIC MODELING PERSPECTIVE

**Equivalent probabilistic formulation for the QAP.** Let $\Delta(\Pi_n) := \{p : \Pi_n \to \mathbb{R} \mid p(\pi) \geq 0, \forall \pi \in \Pi_n; \sum_{\pi \in \Pi_n} p(\pi) = 1\}$ be the probabilistic simplex over the permutation space $\Pi_n$. We consider the optimization problem over this probabilistic space

$$\min_{p \in \Delta(\Pi_n)} \mathbb{E}_{\pi \sim p}[f(\pi; \mathcal{P})] := \sum_{\pi \in \Pi_n} f(\pi; \mathcal{P}) p(\pi \mid \mathcal{P}). \tag{3}$$

This probabilistic formulation is provably equivalent to the original deterministic model (1) of the QAP, as formalized in Theorem 1 with the proof given in Appendix A.

**Theorem 1.** *Let $v^\star = \min_{\pi \in \Pi_n} f(\pi; \mathcal{P})$ and $w^\star = \min_{p \in \Delta(\Pi_n)} \mathbb{E}_p[f(\pi; \mathcal{P})]$. Then $w^\star = v^\star$. Moreover, the minimizers of (3) are precisely the distributions supported on the optimal solution set of (1).*

**From optimization to learning via parametric models.** It is computationally intractable to directly over the entire probability simplex due to the factorial growth of its dimension. In practice, we would replace the whole probabilistic space with a family of probabilistic models $\{p_\theta(\cdot \mid \mathcal{P}) \mid \theta \in \mathbb{R}^d\}$ parametrized by a neural network with parameters $\theta$. To allow learning transferable distribution knowledge, this network is instance-conditioned, which takes a problem instance $\mathcal{P}$ as input and generates a conditional probabilistic distribution $p_\theta(\cdot \mid \mathcal{P})$. The training for this network aims to minimize the expected cost over a distribution of instances $\Gamma$, leading to the training objective

$$\min_{\theta \in \mathbb{R}^d} \mathbb{E}_{\mathcal{P} \sim \Gamma} \mathbb{E}_{\pi \sim p_\theta(\cdot \mid \mathcal{P})}[f(\pi; \mathcal{P})]. \tag{4}$$

**Pushforward transformation.** Directly minimizing this expectation is difficult because the objective landscape is typically rugged and riddled with poor local minima. To address this, we push the entire distribution $p_\theta$ forward to yield a flatter distribution. Let $\mathcal{T} : \Pi_n \to \Pi_n$ be a local improvement map whose specific implementation is detailed in Appendix B.1. The pushforward distribution $\mathcal{T}_\sharp p_\theta$ is then defined as $(\mathcal{T}_\sharp p_\theta)(\sigma) := \sum_{\pi \in \mathcal{T}^{-1}(\sigma)} p_\theta(\pi)$, which aggregates the probability mass of all permutations $\pi$ that are mapped by $\mathcal{T}$ to the same improved solution $\sigma = \mathcal{T}(\pi)$. In this way, it creates a flatter distribution supported on the set of improved solutions, reducing the risk of getting trapped in poor local optima. After applying the pushforward transformation, the objective becomes

$$\min_{\theta \in \mathbb{R}^d} \mathcal{L}(\theta) := \mathbb{E}_{\mathcal{P} \sim \Gamma} \mathbb{E}_{\sigma \sim \mathcal{T}_\sharp p_\theta(\cdot \mid \mathcal{P})}[f(\sigma; \mathcal{P})] = \mathbb{E}_{\mathcal{P} \sim \Gamma} \mathbb{E}_{\pi \sim p_\theta(\cdot \mid \mathcal{P})}[f(\mathcal{T}(\pi); \mathcal{P})]. \tag{5}$$

The second equality above, which follows from the change-of-variables formula for pushforward measures, is pivotal to our method. It shows that the underlying sampler $p_\theta$ is trained to place probability mass on regions that yield high-quality solutions after local improvement. At the same time, it provides a practical way to optimize $\mathcal{L}(\theta)$: one can sample from the original, easy-to-sample distribution $p_\theta$ and then evaluate the improved cost $f(\mathcal{T}(\pi); \mathcal{P})$.

## 3 PLMA FRAMEWORK

### 3.1 LEARNING WITH AN EFFICIENT ENERGY-BASED MODEL

Given a heatmap from the network, existing work (Wang et al., 2019; 2021b) often projects it into permutations using the non-differentiable Hungarian algorithm, which limits diversity and optimization. To circumvent these issues, we introduce a subtly designed energy-based model (EBM) that is computationally tailored and supports efficient exploration of the permutation space via Metropolis–Hastings (MH) sampling. The relationship between a permutation matrix $X_\pi$ and a heatmap $\phi = \phi(\theta, \mathcal{P}) \in \mathbb{R}^{n \times n}$ can be modeled using a EBM, derived from a distance metric as follows:

$$p_\theta(\pi \mid \mathcal{P}) \propto \exp\left(-\frac{1}{2}\|X_\pi - \phi\|_F^2\right) = \exp\left(\langle X_\pi, \phi\rangle - \frac{1}{2}\|X_\pi\|_F^2 - \frac{1}{2}\|\phi\|_F^2\right) \tag{6}$$
$$\propto \exp\left(\langle X_\pi, \phi\rangle\right) =: \exp\left(\Phi_\theta(\pi)\right),$$

where it is simplified because $\|X_\pi\|_F^2$ is a constant value of $n$ and $\|\phi\|_F^2$ is also constant with respect to the permutation matrix $X_\pi$. The resulting additive score, defined as $\Phi_\theta(\pi) := \langle X_\pi, \phi\rangle = \sum_{i=1}^n \phi_{i,\pi(i)}$, is the key to our approach. This additive design is deliberate, as it allows efficient evaluation of score differences when exploring the space of permutations.

To sample from this distribution, which is known only up to the intractable $Z_\theta$, we employ the MH algorithm with proposal kernel $\kappa(\cdot \mid \pi)$ and acceptance probability

$$\alpha(\pi \to \pi') = \min\left\{1, \frac{p_\theta(\pi')\,\kappa(\pi \mid \pi')}{p_\theta(\pi)\,\kappa(\pi' \mid \pi)}\right\}. \tag{7}$$

We adopt a symmetric 2-swap proposal that uniformly selects two positions $a \neq b$ and swaps $\pi(a)$ with $\pi(b)$. The proposal's symmetry $\kappa(\pi \mid \pi') = \kappa(\pi' \mid \pi)$ simplifies the Hastings ratio in (7) to

$$\exp(\Phi_\theta(\pi') - \Phi_\theta(\pi)) = \exp\left(\phi_{a,\pi(b)} + \phi_{b,\pi(a)} - \phi_{a,\pi(a)} - \phi_{b,\pi(b)}\right),$$

which can be computed in constant time. Thus, each update has $O(1)$ cost, and the stationary distribution of the chain remains $p_\theta(\pi)$ with the normalization constant canceled. This leads to a remarkably efficient MCMC sampling scheme.

Although the exact log-likelihood $\log p_\theta(\pi)$ is not available due to the intractable $Z_\theta$, we can still estimate its gradient without computing $Z_\theta$. In particular, an unbiased and consistent Monte Carlo estimator enables the use of policy-gradient optimization. This is formalized in Theorem 2, with the proof provided in Appendix A. We then train the model under the learning objective (5). The specific pretraining algorithm is provided in Appendix B.2.

**Theorem 2.** *Let $\pi_1, ..., \pi_N$ be i.i.d. samples from $p_\theta$ in (6). Then for any function $g : \Pi_n \to \mathbb{R}$, the gradient of the expectation $\mathbb{E}_{\pi \sim p_{\Phi_\theta}}[g(\pi)]$ admits an unbiased and consistent estimator*

$$\hat{G}_N := \frac{1}{N-1}\sum_{i=1}^N \left(g(\pi_i) - \frac{1}{N}\sum_{i=1}^N g(\pi_i)\right)\nabla_\theta \Phi_\theta(\pi_i). \tag{8}$$

### 3.2 GENERATING HEATMAPS WITH A CROSS-GRAPH ATTENTION NETWORK

A QAP instance can be represented as two weighted graphs, one for locations and one for facilities. To maintain scalability, we encode each graph separately with edge-based GNNs, embedding the $n^2$ pairwise information into edges and aggregating it into $n$ node embeddings. A cross-graph attention module then captures interactions between the two graphs, and the resulting embeddings are combined through a dot product and Sinkhorn normalization to form a heatmap $\phi(\theta, \mathcal{P})$.

**Node embedding for each graph.** Initial node features are set by learnable parameters $\boldsymbol{h}_{\text{ini}} \in \mathbb{R}^{d_{in}}$ as $\boldsymbol{X}^0 = \mathbf{1}\boldsymbol{h}_{\text{ini}}^T \in \mathbb{R}^{2n \times d_{in}}$, which are then projected to the initial embeddings $\boldsymbol{H}^{(0)} = \boldsymbol{X}^0\boldsymbol{W}_{\text{proj}} \in \mathbb{R}^{2n \times d}$. These embeddings are subsequently updated through $l_1$ message passing layers. Each layer updates the node embeddings $\boldsymbol{H}$ based on the input graph $\boldsymbol{D}$ and $\boldsymbol{F}$ by combining a standard graph convolution network (GCN) operation with a direct algebraic operation on the weight matrix:

$$\boldsymbol{H}^{(l)} = \begin{bmatrix} \boldsymbol{H}_D^{(l)} \\ \boldsymbol{H}_F^{(l)} \end{bmatrix} = \sigma\left[\begin{bmatrix} \boldsymbol{D} - \bar{\boldsymbol{D}} & \\ & \boldsymbol{F} - \bar{\boldsymbol{F}} \end{bmatrix}\begin{bmatrix} \boldsymbol{H}_D^{(l-1)}\boldsymbol{W}_{\text{alg,D}}^l \\ \boldsymbol{H}_F^{(l-1)}\boldsymbol{W}_{\text{alg,F}}^l \end{bmatrix}\right]. \tag{9}$$

Here, $\bar{\boldsymbol{D}} = \bar{d}\mathbf{1}\mathbf{1}^{\mathrm{T}}$ and $\bar{\boldsymbol{F}} = \bar{f}\mathbf{1}\mathbf{1}^{\mathrm{T}}$, and $\bar{d}, \bar{f}$ are the averages of elements of $\boldsymbol{D}$ and $\boldsymbol{F}$. The layers are stacked with residual connections and layer normalization. By incorporating the edge information into the weighted convolutions, the original $2n^2$ pair information is implicitly transferred into the $2n$ node embeddings while retaining the scalability.

**Cross-graph attention.** To break the disconnectedness of two graphs, a cross-attention module updates the embeddings. This module is analogous to a standard transformer encoder block. It allows each set of embeddings to be updated based on information from the other, producing contextually aware final embeddings. The initial embeddings are set as the GCN output: $\tilde{\boldsymbol{H}}_D^{(0)} = \boldsymbol{H}_D^{(l_1)}, \tilde{\boldsymbol{H}}_F^{(0)} = \boldsymbol{H}_F^{(l_1)}$. In the $r$-th of the total $l_2$ layers, the attention scores $e_{D,ij}^r$ and $e_{F,ij}^r$ are calculated:

$$a_{D,ij}^{(r)} = (\tilde{\boldsymbol{h}}_{D,i}^{r-1}\boldsymbol{W}_{D,q})(\tilde{\boldsymbol{h}}_{F,j}^{r-1}\boldsymbol{W}_{D,k})^{\mathrm{T}}, \quad a_{F,ij}^{(r)} = (\tilde{\boldsymbol{h}}_{F,i}^{r-1}\boldsymbol{W}_{F,q})(\tilde{\boldsymbol{h}}_{D,j}^{r-1}\boldsymbol{W}_{F,k})^{\mathrm{T}},$$

$$e_{D,ij}^r = \frac{\exp(a_{D,ij}^{(r)})}{\sum_{k=1}^n \exp(a_{D,ik}^{(r)})}, \quad e_{F,ij}^r = \frac{\exp(a_{F,ij}^{(r)})}{\sum_{k=1}^n \exp(a_{F,ik}^{(r)})}, \tag{10}$$

where $\tilde{\boldsymbol{h}}_{D,i}^{r-1}$ and $\tilde{\boldsymbol{h}}_{F,i}^{r-1}$ are the $i$-th rows of $\tilde{\boldsymbol{H}}_D^{(r-1)}$ and $\tilde{\boldsymbol{H}}_F^{(r-1)}$. Given the score matrices $\boldsymbol{E}_D^r = (e_{D,ij}^r)$ and $\boldsymbol{E}_F^r = (e_{F,ij}^r)$, the embeddings are updated via aggregating values weighted by attention scores:

$$\tilde{\boldsymbol{H}}^{(r)} = \begin{bmatrix} \tilde{\boldsymbol{H}}_D^{(r)} \\ \tilde{\boldsymbol{H}}_F^{(r)} \end{bmatrix} = \tilde{\boldsymbol{H}}^{(r-1)} + \mathrm{MLP}^r \left[ \begin{bmatrix} & \boldsymbol{E}_D^r \\ \boldsymbol{E}_F^r & \end{bmatrix} \begin{bmatrix} \tilde{\boldsymbol{H}}_D^{(r-1)}\boldsymbol{W}_{D,k}^r \\ \tilde{\boldsymbol{H}}_F^{(r-1)}\boldsymbol{W}_{F,k}^r \end{bmatrix} \right]. \tag{11}$$

Note that to avoid extending the node space to $n^2$, the feature is compressed into the $2n$ nodes and the matrices $\boldsymbol{D}$ and $\boldsymbol{F}$ are inserted into the block-diagonal pattern in (9), leaving the unknown block off-diagonal relationship empty. Therefore, the attention scores $\boldsymbol{E}_D$ and $\boldsymbol{E}_F$ are calculated to reflect the dynamic relationship for the unknown pairs and fill in the block off-diagonal pattern in (11). By leveraging both the block diagonal and off-diagonal relationships, the original two matrices $\boldsymbol{D}$ and $\boldsymbol{F}$ are embedded into two set of node embeddings and their complex relationship is reflected.

**Heatmap generation.** Finally, the network generates a heatmap over facility-location pairs. Concretely, the heatmap is constructed via the dot product between facility and location embeddings, followed by element-wise $\tanh$ clipping and a Sinkhorn normalization in the log-domain:

$$\phi(\theta, \mathcal{P}) = \text{log-Sinkhorn}\left[C\tanh\left(\frac{\tilde{\boldsymbol{H}}_F^{(l_2)}(\tilde{\boldsymbol{H}}_D^{(l_2)})^{\mathrm{T}}}{\sqrt{d}}\right)\right], \tag{12}$$

where $C > 0$ is a clipping constant. The $\tanh$ clipping here increases nonlinearity while preventing extreme logits. The log-Sinkhorn operator pushes the heatmap toward a doubly stochastic matrix in log-domain. This structure actually yields a low-rank approximation of some permutation matrix in the log-domain, as mentioned in (Dröge et al., 2023). Ideally we hope this approximated permutation matrix corresponds to an optimal solution to the problem in the log-domain.

## 3.3 BATCHED WARM-STARTED MCMC FINETUNING

To enhance generalization, we introduce an MCMC-based finetuning stage that efficiently adapts the model to new target instances. As detailed in Algorithm 1, the procedure begins by generating $K$ initial states $\{\pi_{i,k}\}_{k\in[K]}$ for each problem instance $\mathcal{P}_i$ using a long-run MH sampler to ensure high-quality starting points. At each subsequent finetuning step, $M$ parallel Markov chains of length $L$ are launched from every $\pi_{i,k}$, producing terminal states $\{\pi_{i,k}^m\}_{m\in[M]}$. These terminal states are then refined by a local improvement map to $\hat{\pi}_{i,k}^m = \mathcal{T}(\pi_{i,k}^m)$, after which the objectives of these refined samples are evaluated. This entire process of sampling, improvement and evaluation runs in parallel for all instances in a batch. Next, the resulting samples are aggregated to estimate a single policy gradient which is used to perform a synchronized update on the shared model parameters. Finally, the initial states for the next iteration follow a within group best retention rule. For each $k$, the element of $\{\hat{\pi}_{i,k}^m\}_{m\in[M]}$ with the smallest objective value is selected as the next initial state. In this way, high quality solutions from the previous iteration are preserved as warm starts that steer subsequent sampling toward promising regions and promote steady convergence.

Our finetuning is performed in a batch-wise manner, enabling the network to simultaneously explore for high-quality solutions across different instances. In contrast to traditional per-instance search methods that adjust model parameters separately for each test instance, this highly parallel approach enhances computational efficiency and leads to a considerable reduction in runtime. While instances within a batch share a common set of network parameters during the update, our experiment in Appendix D.1 confirms that this shared adaptation scheme does not impede final solution quality.

The efficacy of our finetuning framework hinges on two core design principles. The first is the dynamic use of Markov chain lengths. For generating the initial set of start points, we employ long chains, which ensures the initial points are high-quality samples drawn faithfully from the model to leverage the power of pretraining. Once finetuning commences, we switch to using short chains which facilitate the preservation of structural information carried by the initial states. The second is the mechanism for interplay across chains, which is crucial for striking a trade off between solution quality and sample diversity. If all filtered samples are indiscriminately pooled as starting points for the next iteration, diversity is abundant but instability arises from low quality chains. If only globally best samples are retained, trajectories derived from the same initial state tend to dominate and diversity deteriorates. In contrast, our within group competitive mechanism suppresses the adverse influence of inferior chains while preventing the loss of global diversity.

---

**Algorithm 1** Batched warm-started MCMC finetuning

---

1: **Input:** pretrained EBM $p_\theta$ with score function $\Phi_\theta$, a batch of instances $\{\mathcal{P}_1, \ldots, \mathcal{P}_B\}$, number of starting points $K$, number of chains per sample $M$, length of chains $L$, number of epochs $T$.

2: Use a long-run MH sampler to generate initial points $\pi_{i,k} \sim p_\theta(\cdot \mid \mathcal{P}_i)$. $\quad (i \in [B], k \in [K])$

3: **for** $t = 1, ..., T$ **do**

4:    **for** each instance $i \in [B]$ **in parallel do**

5:       Starting from $\pi_{i,k}$, use the MH sampling with M chains of length $L$ to sample $\{\pi_{i,k}^m\}_{m\in[M]}$ from distribution $p_\theta(\cdot \mid \mathcal{P}_i)$. $\quad (k \in [K])$

6:       Apply local improvement map to obtain $\hat{\pi}_{i,k}^m = \mathcal{T}(\pi_{i,k}^m)$. $\quad (k \in [K], m \in [M])$

7:       Evaluate objective for improved samples: $\hat{f}_{i,k}^m = f(\hat{\pi}_{i,k}^m; \mathcal{P}_i)$. $\quad (k \in [K], m \in [M])$

8:    **end for**

9:    $b_i \leftarrow \frac{1}{KM} \sum_{k=1}^{K} \sum_{m=1}^{M} \hat{f}_{i,k}^m, \, i \in [B], \quad \hat{G} \leftarrow \frac{1}{B(KM-1)} \sum_{i=1}^{B} \sum_{k=1}^{K} \sum_{m=1}^{M} (\hat{f}_{i,k}^m - b_i) \nabla_\theta \Phi_\theta(\pi_{i,k}^m).$

10:    $\theta \leftarrow \text{Adam}(\theta, \hat{G}).$

11:    $\pi_{i,k} \leftarrow \arg\min\{ f(\hat{\pi}) \mid \hat{\pi} \in \{\hat{\pi}_{i,k}^m\}_{m\in[M]} \}.$ $\quad (i \in [B], k \in [K])$

12: **end for**

---

# 4 EXPERIMENTS

## 4.1 EXPERIMENTAL SETUP

**Datasets.** We evaluate our model on two families of synthetic QAP instances and on the real world QAPLIB (Burkard et al., 1997) and Taixxeyy (Drezner et al., 2005) benchmarks. The synthetic data comprise geometrically structured instances generated as in (Tan & Mu, 2024) and uniformly random instances produced following (Taillard, 1991). The main experiments use sizes $n \in \{20, 50, 100\}$, and for each size, we construct 256 instances per family. Scalability is assessed on large-scale instances with $n \in \{200, 500\}$, where we create 64 instances for each large size, and all of these large instances are uniformly random. A complete description of these datasets can be found in Appendix B.3.

**Baselines.** We compare our model against a wide spectrum of established and modern approaches, categorized as follows: (i) For **search-based solvers**, we include a highly-optimized heuristic solver, Robust Tabu Search (Ro-TS) (Taillard, 1991); (ii) For **heuristic solvers**, we benchmark against classic iterative algorithms often used for graph matching, including IPFP (Leordeanu et al., 2009), SM (Leordeanu & Hebert, 2005), and RRWM (Cho et al., 2010). (iii) For **learning-based solvers**, we include two recent deep learning approaches, SAWT (Tan & Mu, 2024) and NGM (Wang et al., 2021b). The implementation details for these baselines are available in Appendix B.4.

Table 1: Performance comparison on synthetic datasets (256 instances). Ro-TS (xk) denotes Ro-TS with x*1000*n iterations. IPFP (x) runs IPFP from x random initializations. PLMA ($T = 1$) represents the zero-shot performance of the pre-trained model without any fine-tuning, while $T = 50/200$ correspond to the results after 50 / 200 fine-tuning iterations.

| Algorithm | QAP20 | | | QAP50 | | | QAP100 | | |
| --- | --- | --- | --- | --- | --- | --- | --- | --- | --- |
| | Cost | Gap | Time | Cost | Gap | Time | Cost | Gap | Time |
| *Geometrically Structured Datasets* | | | | | | | | | |
| Ro-TS (1k) | 54.38 | 0.01% | 17.04s | 375.99 | 0.14% | 4m35s | 1593.27 | 0.13% | 38m56s |
| Ro-TS (5k) | **54.37** | **0.00%** | 1m25s | **375.48** | **0.00%** | 22m53s | 1591.25 | 0.00% | 3h15m |
| IPFP | 55.11 | 1.37% | 2.04s | 378.76 | 0.88% | 11.47s | 1600.27 | 0.57% | 1m34s |
| IPFP (10) | 54.54 | 0.31% | 20.97s | 376.60 | 0.30% | 2m30s | 1594.76 | 0.22% | 17m34s |
| RRWM | 71.30 | 31.09% | 11.30s | 428.78 | 14.14% | 39.23s | 1700.33 | 6.86% | 6m32s |
| SM | 64.38 | 18.45% | 0.22s | 426.92 | 13.70% | 7.14s | 1753.10 | 10.17% | 1m40s |
| NGM | 62.93 | 15.87% | 24.78s | 429.69 | 14.46% | 1m16s | 1773.71 | 11.47% | 2m29s |
| SAWT (10k) | 54.72 | 0.64% | 3m41s | 380.92 | 1.45% | 5m36s | 1617.30 | 1.64% | 10m43s |
| **PLMA** ($T = 1$) | 54.63 | 0.48% | 0.06s | 379.79 | 1.15% | 0.41s | 1607.84 | 1.04% | 4.27s |
| **PLMA** ($T = 50$) | **54.37** | **0.00%** | 2.57s | 375.55 | 0.20% | 19.88s | 1591.73 | 0.03% | 3m30s |
| **PLMA** ($T = 200$) | **54.37** | **0.00%** | 9.36s | **375.48** | **0.00%** | 1m19s | **1591.23** | **0.00%** | 13m58s |
| *Uniformly Random Datasets* | | | | | | | | | |
| Ro-TS(1k) | 76.61 | 0.07% | 17.56s | 523.08 | 0.22% | 4m36s | 2195.98 | 0.13% | 38m59s |
| Ro-TS(5k) | **76.56** | **0.00%** | 1m27s | 521.91 | 0.00% | 22m59s | 2193.16 | 0.00% | 3h15m |
| IPFP | 79.13 | 3.39% | 2.12s | 530.74 | 1.69% | 7.45s | 2211.38 | 0.83% | 41.95s |
| IPFP(25) | 77.60 | 1.37% | 54.70s | 526.96 | 0.97% | 4m20s | 2203.29 | 0.46% | 19m20s |
| RRWM | 93.50 | 22.27% | 11.11s | 592.50 | 13.54% | 31.91s | 2432.34 | 10.91% | 5m6s |
| SM | 92.32 | 20.73% | 0.19s | 605.08 | 15.95% | 5.65s | 2457.47 | 12.05% | 1m23s |
| NGM | 88.34 | 15.49% | 25.08s | 594.99 | 14.01% | 1m17s | 2438.52 | 11.19% | 2m29s |
| **PLMA** ($T = 1$) | 78.23 | 2.20% | 0.06s | 538.42 | 3.17% | 0.40s | 2243.43 | 2.29% | 4.32s |
| **PLMA** ($T = 50$) | 76.59 | 0.05% | 2.47s | 523.95 | 0.40% | 19.51s | 2200.40 | 0.34% | 3m30s |
| **PLMA** ($T = 200$) | **76.56** | **0.00%** | 9.60s | **521.83** | **-0.01%** | 1m18s | **2193.13** | **0.00%** | 14m1s |

**Metrics.** We primarily evaluate all methods using three metrics: (i) Cost, the average objective value over all test instances (lower is better); (ii) Gap, the average relative gap from a reference solution $C_{\text{ref}}$, computed as $(C_{\text{alg}} - C_{\text{ref}})/C_{\text{ref}} \times 100\%$; and (iii) Time, the total computation time to solve the entire test set.

## 4.2 RESULTS

**Results on synthetic QAP datasets.** The results in Table 1 establish PLMA as the new state-of-the-art. With 200 finetuning steps, PLMA consistently matches or surpasses the strong Ro-TS (5k) baseline in quality while being remarkably more efficient; on challenging QAP100 instances, it achieves the same 0.00% optimality gap over 13 times faster. The results also validate our two-stage design. The pre-trained model (PLMA, $T = 1$) offers an instant, high-quality solution that already outperforms other learning-based methods (NGM, SAWT). With brief fine-tuning (PLMA, $T = 50$), the model rapidly converges to near-optimality, demonstrating the synergy between a strong pre-trained starting distribution and precise MCMC-based refinement. See Figure 4 in Appendix D.3 for convergence curves.

**Results on large-scale instances.** To assess the scalability of the PLMA framework, we evaluate its performance on instances of a significantly larger scale ($n \in \{200, 500\}$). The results in Table 2 confirm that our approach scales effectively to these larger problem sizes. Remarkably, The performance of PLMA is competitive with or even superior to Ro-TS, but achieved at a fraction of the computational cost.

**Results on QAPLIB.** We evaluate PLMA on the widely-used QAPLIB benchmark, with results summarized in Table 3. PLMA achieves the best overall solution quality, obtaining an average opti-

Table 2: Performance comparison on large-scale synthetic datasets (64 instances).

| Algorithm | QAP200 | | | QAP500 | | |
|---|---|---|---|---|---|---|
| | Cost | Gap | Time | Cost | Gap | Time |
| Ro-TS | 6640.28 | 0.00% | 3h48m | 43028.56 | 0.00% | 29h59m |
| SAWT (10k) | 6735.97 | 1.44% | 8m33s | 43696.84 | 1.55% | 48m36s |
| **PLMA** ($T = 1$) | 6723.41 | 1.25% | 8.02s | 43626.09 | 1.39% | 29.89s |
| **PLMA** ($T = 50$) | 6644.45 | 0.06% | 6m44s | 43035.81 | 0.02% | 24m54s |
| **PLMA** ($T = 200$) | **6639.13** | **-0.02%** | 26m44s | **43022.97** | **-0.06%** | 1h40m |

Table 3: Performance on QAPLIB. Average gaps and computation time (s) are reported for each category. The search terminates upon finding the optimal solution or reaching the iteration limit.

| Instance Class | Ro-TS | | SAWT | | IPFP | | PLMA | |
|---|---|---|---|---|---|---|---|---|
| | Gap | Time | Gap | Time | Gap | Time | Gap | Time |
| bur (26) | **0.00%** | 0.12 | 3.95% | 14.67 | 0.05% | 0.33 | **0.00%** | 0.07 |
| chr (12-25) | 0.48% | 0.16 | 147.54% | 14.18 | 14.90% | 0.26 | **0.20%** | 0.94 |
| els (19) | **0.00%** | 0.02 | 47.37% | 14.24 | 10.18% | 0.36 | **0.00%** | 0.08 |
| esc (16-128) | **0.00%** | 0.62 | 43.29% | 15.07 | 0.39% | 0.63 | **0.00%** | 0.04 |
| had (12-20) | **0.00%** | 0.00 | 5.17% | 14.23 | 0.08% | 0.39 | **0.00%** | 0.02 |
| kra (30-32) | **0.00%** | 0.24 | 32.92% | 14.77 | 0.65% | 0.62 | **0.00%** | 0.77 |
| lipa (20-90) | **0.03%** | 3.28 | 1.40% | 17.32 | 1.07% | 4.93 | 0.10% | 1.77 |
| nug (12-30) | **0.00%** | 0.02 | 19.25% | 14.39 | 0.05% | 0.38 | **0.00%** | 0.14 |
| rou (12-20) | **0.00%** | 0.04 | 15.09% | 14.25 | 0.77% | 0.33 | 0.02% | 1.77 |
| scr (12-20) | **0.00%** | 0.01 | 33.92% | 14.22 | 1.24% | 0.28 | **0.00%** | 0.04 |
| sko (42-100) | 0.05% | 29.22 | 16.17% | 19.06 | 0.30% | 8.40 | **0.03%** | 7.10 |
| ste (36) | **0.01%** | 0.53 | 107.95% | 14.95 | 1.81% | 0.70 | **0.01%** | 0.77 |
| tai (10-256) | **0.23%** | 25.16 | 34.67% | 16.62 | 0.92% | 3.33 | 0.28% | 4.93 |
| tho (30-150) | **0.04%** | 38.74 | 24.05% | 17.19 | 0.42% | 12.46 | 0.11% | 5.29 |
| wil (50-100) | **0.02%** | 26.06 | 9.52% | 17.64 | 0.12% | 9.66 | **0.02%** | 5.48 |
| **Average** | 0.11% | 9.68 | 37.82% | 15.85 | 2.15% | 2.73 | **0.10%** | 2.29 |

mality gap of 0.10% while being over four times faster than the strong Ro-TS baseline. PLMA shows particular strength on some specific categories like "chr" and "sko". This performance demonstrates a superior overall balance of solution quality, efficiency, and generalization.

### 4.3 ABLATION STUDY

We conduct ablation studies on the uniformly random dataset ($n = 100$) to isolate and verify the contributions of PLMA's core components, with results shown in Figure 1.

**Analysis of network architecture components.** We assess our network design by individually removing the cross-attention module (No Cross-att) and the Sinkhorn normalization layer (No Sinkhorn). The results indicate that the absence of either component leads to a distinct degradation in solution quality, with the removal of Sinkhorn normalization causing a more substantial performance drop (Figure 1a). This confirms the synergistic and essential roles these components play in fusing inter-graph information and structuring the probabilistic search space.

**Effectiveness of warm-started MCMC finetuning.** In Figure 1b, we compare our finetuning strategy against two alternatives. AR-Seq uses an auto-regressive model (AR) and generates solutions by sequentially constructive sampling (details in Appendix B.4). GD-Free performs MCMC search without policy gradient updates. The results reveal two key advantages of our approach. First, it overcomes the restart-from-scratch limitation of AR models by enabling warm-starts from prior solutions. Second, its superiority over the GD-Free variant confirms that gradient-based policy updates are crucial for achieving state-of-the-art precision. Further hyperparameter analysis is provided in Appendix D.2.

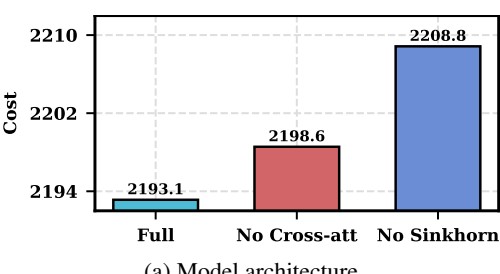
(a) Model architecture

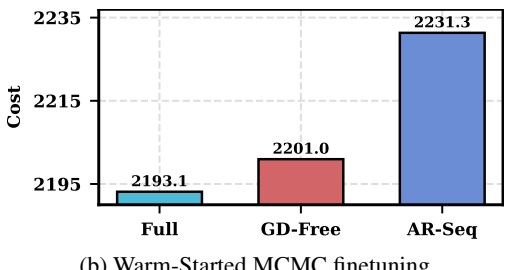
(b) Warm-Started MCMC finetuning

Figure 1: Ablations on the uniformly random dataset ($n = 100$).

Table 4: Grouped results on Taixxeyy instances, with all metrics averaged over 10 independent runs. For each class, the reported mean and [min, max] gaps (%) are averages of per-instance statistics.

| Class | Ro-TS | | | IPFP | | | PLMA | | |
|---|---|---|---|---|---|---|---|---|---|
| | mean | [min, max] | Time (s) | mean | [min, max] | Time (s) | mean | [min, max] | Time (s) |
| tai27e | 41.50 | [0.11, 221.08] | 0.57 | 19.47 | [6.14, 34.11] | 0.38 | **0.00** | [0.00, 0.00] | 0.08 |
| tai45e | 101.89 | [1.00, 400.60] | 3.83 | 22.01 | [8.26, 36.98] | 1.03 | **0.03** | [0.00, 0.35] | 0.28 |
| tai75e | 111.28 | [6.20, 280.01] | 18.49 | 27.64 | [17.56, 36.78] | 2.52 | **0.09** | [0.00, 0.45] | 1.48 |
| tai125e | 82.53 | [7.65, 265.54] | 72.52 | 26.93 | [20.64, 32.67] | 8.52 | **2.67** | [-0.08, 5.62] | 8.18 |
| tai175e | 67.86 | [9.11, 260.98] | 158.18 | 23.32 | [17.70, 28.11] | 16.86 | **9.11** | [5.61, 12.04] | 14.63 |
| Average | 81.01 | [4.82, 285.64] | 50.72 | 23.87 | [14.06, 33.73] | 5.86 | **2.38** | [1.11, 3.69] | 4.93 |

## 4.4 ROBUSTNESS ANALYSIS ON TAIXXEYY INSTANCES

We test robustness on challenging Taixxeyy instances over 10 independent runs. In Table 4, we report grouped results. PLMA delivers a low average gap of 2.38% and a reliable worst-case performance with an average maximum gap of 3.69%. This is in stark contrast to Ro-TS's highly erratic 81.01% average gap and catastrophic failures with a maximum gap of 285.64%. To further assess robustness, we visualize the distribution of final gaps in Figure 2. Outliers with gap greater than 100% are plotted as individual points, and the boxplots are drawn after removing these outliers. In contrast to RoTS, which shows frequent extreme failures, PLMA yields no outliers and its boxes are distinctly narrower, indicating stable behavior across trials.

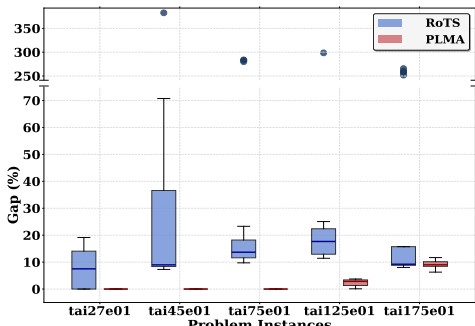

Figure 2: Gap distributions on representative Taixxeyy instances (10 runs).

## 5 CONCLUSION

This paper presents PLMA, a permutation learning framework for the QAP that narrows the gap between neural methods and strong handcrafted heuristics. The finetuning stage exploits the warm start behavior of short MCMC chains so that each iteration of finetuning stays near previously discovered high quality assignments instead of restarting from scratch. A meticulously designed energy-based model with additive structure enables rapid exploration of the permutation space through constant time swap evaluations. The cross-graph attention mechanism offers a scalable and flexible way to model the intertwined two-graph structure of the QAP. In extensive experiments, PLMA shows state-of-the-art performance across synthetic and real-world benchmarks. To the best of our knowledge, PLMA is the first learning-based approach to achieve a near zero average gap on QAPLIB while running more than four times faster than Ro-TS and showing stronger robustness on difficult Taixxeyy instances. Future work will resort to meta learning to better bridge pretraining and finetuning, and will extend the framework to broader classes of permutation-based problems.

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

# A PROOFS

**Theorem 1.** *Let $v^\star = \min\limits_{\pi \in \Pi_n} f(\pi; \mathcal{P})$ and $w^\star = \min\limits_{p \in \Delta(\Pi_n)} \mathbb{E}_p[f(\pi; \mathcal{P})]$. Then $w^\star = v^\star$. Moreover, the minimizers of (3) are precisely the distributions supported on the optimal solution set of (1).*

*Proof.* For any $p \in \Delta(\Pi_n)$, $\mathbb{E}_p[f] \geq \min_\pi f(\pi) = v^\star$, hence $w^\star \geq v^\star$. If $\pi^\star \in \arg\min_\pi f(\pi)$ and $p = \delta_{\pi^\star}$, then $\mathbb{E}_p[f] = f(\pi^\star) = v^\star$, hence $w^\star \leq v^\star$. Thus $w^\star = v^\star$. Linearity on the simplex implies optima occur at extreme points (Dirac measures) or their convex combinations over the minimizer set. $\qquad\square$

**Theorem 2.** *Let $\pi_1, ..., \pi_N$ be i.i.d. samples from $p_\theta$ in (6). Then for any function $g : \Pi_n \to \mathbb{R}$, the gradient of the expectation $\mathbb{E}_{\pi \sim p_{\Phi_\theta}}[g(\pi)]$ admits an unbiased and consistent estimator*

$$\hat{G}_N := \frac{1}{N-1} \sum_{i=1}^{N} \left( g(\pi_i) - \frac{1}{N} \sum_{i=1}^{N} g(\pi_i) \right) \nabla_\theta \Phi_\theta(\pi_i). \tag{8}$$

*Proof.* We start by translating the target expectation to a convenient form. Recall that

$$p_\theta(\pi) = \frac{\exp(\Phi_\theta(\pi))}{Z_\theta}, \qquad Z_\theta = \sum_{\hat{\pi} \in \Pi_n} \exp(\Phi_\theta(\hat{\pi})),$$

so we have $\log p_\theta(\pi) = \Phi_\theta(\pi) - \log Z_\theta$. Then for any $\pi$ drawn from $p_\theta$, we have

$$
\begin{aligned}
\mathbb{E}_{\pi \sim p_\theta}\big[g(\pi)\, \nabla_\theta \log p_\theta(\pi)\big] &= \mathbb{E}_{\pi \sim p_\theta}\Big[g(\pi)\left(\nabla_\theta \Phi_\theta(\pi) - \nabla_\theta \log Z_\theta\right)\Big] \\
&= \mathbb{E}_{\pi \sim p_\theta}\Big[g(\pi)\, \nabla_\theta \Phi_\theta(\pi)\Big] - \nabla_\theta \log Z_\theta\, \mathbb{E}_{\pi \sim p_\theta}[g(\pi)] \\
&= \mathbb{E}_{\pi \sim p_\theta}\Big[g(\pi)\, \nabla_\theta \Phi_\theta(\pi)\Big] - \frac{1}{Z_\theta} \sum_{\hat{\pi} \in \Pi_n} \nabla_\theta \exp(\Phi_\theta(\hat{\pi}))\, \mathbb{E}_{\pi \sim p_\theta}[g(\pi)] \\
&= \mathbb{E}_{\pi \sim p_\theta}\Big[g(\pi)\, \nabla_\theta \Phi_\theta(\pi)\Big] - \sum_{\hat{\pi} \in \Pi_n} \frac{\exp(\Phi_\theta(\hat{\pi}))}{Z_\theta} \nabla_\theta \Phi_\theta(\hat{\pi})\, \mathbb{E}_{\pi \sim p_\theta}[g(\pi)] \\
&= \mathbb{E}_{\pi \sim p_\theta}\Big[g(\pi)\, \nabla_\theta \Phi_\theta(\pi)\Big] - \mathbb{E}_{\pi \sim p_\theta}\big[\nabla_\theta \Phi_\theta(\pi)\big]\, \mathbb{E}_{\pi \sim p_\theta}[g(\pi)] \\
&= \mathbb{E}_{\pi \sim p_\theta}\Big[\big(g(\pi) - \mathbb{E}_{\hat{\pi} \sim p_\theta}[g(\hat{\pi})]\big) \nabla_\theta \Phi_\theta(\pi)\Big],
\end{aligned}
$$

where in the second line the constant $\nabla_\theta \log Z_\theta$ is pulled out. The final expression shows that

$$\mathbb{E}_{\pi \sim p_\theta}\big[g(\pi)\,\nabla_\theta \log p_\theta(\pi)\big] \;=\; \mathbb{E}_{\pi \sim p_\theta}\Big[(g(\pi) - \mathbb{E}_{p_\theta}[g(\pi)])\,\nabla_\theta \Phi_\theta(\pi)\Big]\,.$$

**(Unbiasedness)** Now let $\hat{G}_N$ denote the proposed estimator (8) based on $N$ i.i.d. samples. Writing it out:

$$\hat{G}_N \;:=\; \frac{1}{N-1}\sum_{i=1}^{N}\Big(g(\pi_i) - \frac{1}{N}\sum_{j=1}^{N} g(\pi_j)\Big)\,\nabla_\theta \Phi_\theta(\pi_i)\,,$$

where $\pi_1, \ldots, \pi_N \overset{\text{i.i.d.}}{\sim} p_\theta$. We will show $\mathbb{E}[\hat{G}_N]$ equals the target quantity above for any $N$, which establishes unbiasedness.

Using linearity of expectation and the independence of samples, we can expand $\mathbb{E}[\hat{G}_N]$ as follows:

$$\mathbb{E}[\hat{G}_N] = \frac{1}{N-1}\sum_{i=1}^{N}\mathbb{E}\Big[\big(g(\pi_i) - \frac{1}{N}\sum_{j=1}^{N} g(\pi_j)\big)\,\nabla_\theta \Phi_\theta(\pi_i)\Big]$$

$$= \frac{1}{N-1}\sum_{i=1}^{N}\Big(\mathbb{E}[g(\pi_i)\,\nabla_\theta \Phi_\theta(\pi_i)] \;-\; \mathbb{E}\Big[\frac{1}{N}\sum_{j=1}^{N} g(\pi_j)\,\nabla_\theta \Phi_\theta(\pi_i)\Big]\Big)\,.$$

For each fixed $i$, we evaluate the two expectations inside the sum:

- $\mathbb{E}[g(\pi_i)\,\nabla_\theta \Phi_\theta(\pi_i)] = \mathbb{E}_{\pi \sim p_\theta}[g(\pi)\,\nabla_\theta \Phi_\theta(\pi)]$, since $\pi_i$ has the same distribution as a fresh draw $\pi \sim p_\theta$.

- $\mathbb{E}\Big[\frac{1}{N}\sum_{j=1}^{N} g(\pi_j)\,\nabla_\theta \Phi_\theta(\pi_i)\Big] = \frac{1}{N}\sum_{j=1}^{N}\mathbb{E}[\,g(\pi_j)\,\nabla_\theta \Phi_\theta(\pi_i)\,]$
  $= \frac{1}{N}\Big(\mathbb{E}[g(\pi_i)\,\nabla_\theta \Phi_\theta(\pi_i)] + \sum_{j \neq i}\mathbb{E}[g(\pi_j)\,\nabla_\theta \Phi_\theta(\pi_i)]\Big).$

For $j \neq i$, $\pi_j$ is independent of $\pi_i$, so $\mathbb{E}[g(\pi_j)\,\nabla_\theta \Phi_\theta(\pi_i)] \;=\; \mathbb{E}[g(\pi_j)]\,\mathbb{E}[\nabla_\theta \Phi_\theta(\pi_i)] \;=\; \mathbb{E}_{p_\theta}[g(\pi)]\,\mathbb{E}_{p_\theta}[\nabla_\theta \Phi_\theta(\pi)]$. Therefore, continuing the calculation:

$$\mathbb{E}[\hat{G}_N] = \frac{1}{N-1}\sum_{i=1}^{N}\Big(\mathbb{E}[g(\pi)\,\nabla_\theta \Phi_\theta(\pi)] \;-\; \frac{1}{N}\Big[\mathbb{E}[g(\pi)\,\nabla_\theta \Phi_\theta(\pi)] + (N-1)\,\mathbb{E}[g(\pi)]\,\mathbb{E}[\nabla_\theta \Phi_\theta(\pi)]\Big]\Big)$$

$$= \frac{1}{N-1}\sum_{i=1}^{N}\Big(\mathbb{E}[g(\pi)\,\nabla_\theta \Phi_\theta(\pi)] - \frac{1}{N}\mathbb{E}[g(\pi)\,\nabla_\theta \Phi_\theta(\pi)] - \frac{N-1}{N}\,\mathbb{E}[g(\pi)]\,\mathbb{E}[\nabla_\theta \Phi_\theta(\pi)]\Big)$$

$$= \frac{1}{N-1}\sum_{i=1}^{N}\frac{N-1}{N}\Big(\mathbb{E}[g(\pi)\,\nabla_\theta \Phi_\theta(\pi)] - \mathbb{E}[g(\pi)]\,\mathbb{E}[\nabla_\theta \Phi_\theta(\pi)]\Big)$$

$$= \mathbb{E}[g(\pi)\,\nabla_\theta \Phi_\theta(\pi)] \;-\; \mathbb{E}[g(\pi)]\,\mathbb{E}[\nabla_\theta \Phi_\theta(\pi)]$$

$$= \mathbb{E}_{\pi \sim p_\theta}\Big[(g(\pi) - \mathbb{E}_{p_\theta}[g(\pi)])\,\nabla_\theta \Phi_\theta(\pi)\Big]\,.$$

Recall that this quantity is exactly $\mathbb{E}_{\pi \sim p_\theta}[\,g(\pi)\,\nabla_\theta \log p_\theta(\pi)\,]$. Therefore $\mathbb{E}[\hat{G}_N]$ equals the desired one. In other words, $\hat{G}_N$ is an *unbiased* estimator of $\mathbb{E}_{\pi \sim p_\theta}[g(\pi)\,\nabla_\theta \log p_\theta(\pi)]$.

**(Consistency)** Finally, we show that $\hat{G}_N$ converges to the true value as $N \to \infty$. Since $\Pi_n$ is a finite set, $g$ and $\Phi_\theta$ are obviously bounded and then $\mathbb{E}|g(\pi)| < +\infty$, $\mathbb{E}|\nabla_\theta \Phi_\theta(\pi)| < +\infty$ and $\mathbb{E}|g(\pi)\nabla_\theta \Phi_\theta(\pi)| < +\infty$. By the Strong Law of Large Numbers (SLLN), the sample averages converge to their expectations almost surely. In particular, as $N \to \infty$ we have

$$\frac{1}{N}\sum_{i=1}^{N} g(\pi_i) \xrightarrow{\text{a.s.}} \mathbb{E}_{p_\theta}[g(\pi)], \qquad \frac{1}{N}\sum_{i=1}^{N}\nabla_\theta \Phi_\theta(\pi_i) \xrightarrow{\text{a.s.}} \mathbb{E}_{p_\theta}[\nabla_\theta \Phi_\theta(\pi)],$$

and

$$\frac{1}{N}\sum_{i=1}^{N} g(\pi_i)\,\nabla_\theta \Phi_\theta(\pi_i) \xrightarrow{\text{a.s.}} \mathbb{E}_{p_\theta}\left[\,g(\pi)\,\nabla_\theta \Phi_\theta(\pi)\,\right],$$

provided the assumed moment conditions hold. Notice that we can rewrite the estimator in a form convenient for taking the limit:

$$\hat{G}_N \;=\; \frac{N}{N-1}\left(\frac{1}{N}\sum_{i=1}^{N} g(\pi_i)\,\nabla_\theta \Phi_\theta(\pi_i) \;-\; \Big(\frac{1}{N}\sum_{i=1}^{N} g(\pi_i)\Big)\Big(\frac{1}{N}\sum_{i=1}^{N} \nabla_\theta \Phi_\theta(\pi_i)\Big)\right).$$

As $N \to \infty$, the prefactor $\frac{N}{N-1} \to 1$, and each of the big parentheses converges to the corresponding expectation. Hence, by the above limits and continuous mapping, $\hat{G}_N$ converges almost surely to

$$\mathbb{E}_{p_\theta}\left[\,g(\pi)\,\nabla_\theta \Phi_\theta(\pi)\,\right] - \mathbb{E}_{p_\theta}[g(\pi)]\,\mathbb{E}_{p_\theta}[\nabla_\theta \Phi_\theta(\pi)] \;=\; \mathbb{E}_{p_\theta}\left[\,g(\pi)\,\nabla_\theta \log p_\theta(\pi)\,\right],$$

which is the true parameter value we seek. This shows that $\hat{G}_N$ is a *consistent* estimator.

In summary, $\hat{G}_N$ is unbiased and $\hat{G}_N \to \mathbb{E}_{p_\theta}[g(\pi)\,\nabla_\theta \log p_\theta(\pi)]$ almost surely as $N \to \infty$. $\qquad\square$

## B  IMPLEMENTATION DETAILS

### B.1  LOCAL SEARCH FOR QAP

We employ an efficient local search heuristic based on the 2-swap neighborhood. This is a common improvement strategy for many combinatorial optimization problems, including the QAP.

A 2-swap action, denoted as an unordered pair $a = (r, s)$ with $r \neq s$, transforms a permutation $\pi$ into a new permutation $\pi'$ by exchanging the assignments at positions $r$ and $s$:

$$\pi'(i) = \pi(i), \quad \forall i \notin \{r, s\}$$
$$\pi'(r) = \pi(s)$$
$$\pi'(s) = \pi(r)$$

A key advantage of the 2-swap is that the change in the objective function, $\Delta(\pi, a) = f(\pi') - f(\pi)$, can be calculated efficiently without re-evaluating the entire summation. The update can be computed in $O(n)$ **time** as follows:

$$\Delta(\pi, a) = (F_{rr} - F_{ss})(D_{\pi(s)\pi(s)} - D_{\pi(r)\pi(r)}) + (F_{rs} - F_{sr})(D_{\pi(s)\pi(r)} - D_{\pi(r)\pi(s)})$$
$$+ \sum_{k \notin \{r,s\}} \left[ (F_{rk} - F_{sk})(D_{\pi(s)\pi(k)} - D_{\pi(r)\pi(k)}) + (F_{kr} - F_{ks})(D_{\pi(k)\pi(s)} - D_{\pi(k)\pi(r)}) \right]$$

Our local search procedure operates by sampling a batch of potential 2-swap actions in parallel at each iteration. It then identifies the single best action from this batch—the one that yields the greatest decrease in cost. If this best action results in an improved solution, the permutation $\pi$ is updated. This "best-improvement" strategy is detailed in Algorithm 2.

---

**Algorithm 2** Local Search for QAP

---

1: **Input:** Initial solution $\pi$, number of local search iterations $T_{\text{LS}}$, number of 2-swap candidates per iteration $K_{\text{LS}}$.
2: **for** $t = 1, \ldots, T_{\text{LS}}$ **do**
3:     Sample a set of $K_{\text{LS}}$ random 2-swap actions in parallel: $A = \{a_1, a_2, \ldots, a_{K_{\text{LS}}}\}$.
4:     Calculate $\Delta_i = \Delta(\pi, a_i)$ for all $a_i \in A$.
5:     Find the best action: $a^* = \arg\min_{a_i \in A} \Delta_i$.
6:     Let $\Delta^* = \Delta(\pi, a^*)$.
7:     **if** $\Delta^* < 0$ **then**
8:         Update $\pi \leftarrow \pi'$, where $\pi'$ is the result of applying action $a^*$ to $\pi$.
9:     **end if**
10: **end for**
11: **Return:** The refined solution $\pi$.

---

## B.2 TRAINING PROCESS

---

**Algorithm 3** Pre-training with Pushforward Transformation

---

1: **Input:** data distribution $\Gamma$, EBM $p_\theta$ with score function $\Phi_\theta$, number of training steps $S$, batch size B, number of samples $K$ per instance, length of chains $L$.
2: **for** $s = 1, ..., S$ **do**
3:     Sample a batch of instances $\{\mathcal{P}_i\}_{i \in [B]}$ from $\Gamma$.
4:     Use the MH sampler with $N$ chains of length $L$ to sample $\{\pi_{i,k}\}_{k \in [K]}$ from $p_\theta(\cdot \mid \mathcal{P}_i)$ for each instance $\mathcal{P}_i$, where $i = 1, \ldots, B$.
5:     Apply local improvement map to obtain $\hat{\pi}_{i,k} = \mathcal{T}(\pi_{i,k})$.           $(i \in [B], k \in [K])$
6:     Evaluate objective for improved samples: $\hat{f}_{i,k} = f(\hat{\pi}_{i,k})$.           $(i \in [B], k \in [K])$
7:     $b_i \leftarrow \frac{1}{K} \sum_{k=1}^{K} \hat{f}_{i,k}, \, i \in [B], \quad \hat{G} \leftarrow \frac{1}{B(K-1)} \sum_{i=1}^{B} \sum_{k=1}^{K} (\hat{f}_{i,k} - b_i) \nabla_\theta \Phi_\theta(\pi_{i,k})$.
8:     $\theta \leftarrow \mathrm{Adam}(\theta, \hat{G})$.
9: **end for**

---

## B.3 DATASETS DETAILS

An instance of the Quadratic Assignment Problem (QAP) is defined by two $n \times n$ matrices: a distance matrix $D$ and a flow matrix $F$. We use both synthetic datasets and widely-used real-world benchmarks to thoroughly evaluate our model.

**Synthetic Datasets** We generate two distinct classes of synthetic instances to assess performance on both structured and unstructured problems.

- **Geometrically Structured instances (SAWT style):** Following the generation procedure from SAWT(Tan & Mu, 2024), we create instances with geometric structure. The distance matrix $D$ is a 2D-Euclidean distance matrix derived from $n$ coordinates sampled uniformly from the unit square $[0,1]^2$. The flow matrix $F$ is initially sampled from $\mathcal{U}[0,1]$, made symmetric, and then sparsified by randomly setting 70% of off-diagonal elements to zero.

- **Uniformly Random Instances (Tai-a style):** To assess performance on unstructured problems, we generate instances following the 'Tai-a' specification from QAPLIB. The elements of both $D$ and $F$ are sampled independently and identically distributed (i.i.d.) from a uniform distribution $\mathcal{U}[0,1]$ and are then made symmetric.

**Real-World Benchmarks** We also test our model on two widely used and challenging benchmarks, QAPLIB and Taixxeyy.

- **QAPLIB:** QAPLIB (Burkard et al., 1997) is a widely used benchmark for the Quadratic Assignment Problem, comprising 134 real-world QAP instances from 15 categories. Each instance is uniquely named with the format "author name - probelm size - index(optional)".

- **Taixxeyy:** The Taixxeyy benchmark (Drezner et al., 2005) comprises a suite of QAP instances where 'xx' in the name denotes the problem size and 'yy' the instance number. These instances are specifically engineered to be difficult for heuristics that rely on transposition-based local search. They are generated with a recursive, hierarchical block structure, creating clusters of facilities with high intra-group flows and small intra-group distances, but low flows and large distances between groups. To make the problem challenging, this block structure is intentionally obscured with small inter-block flows and large finite distances. This creates a highly rugged solution landscape where meaningful improvement often requires accepting a series of deteriorating solutions.

## B.4 BASELINE DETAILS

We evaluate our method against a range of baselines, from classic heuristics to modern learning-based approaches. We first distinguish between the two primary formulations of the Quadratic Assignment Problem (QAP). The general form, Lawler's QAP, is defined by a quadratic matrix $K$

with the following objective:

$$\min_X \quad \text{vec}(X)^\top K \, \text{vec}(X),$$

$$\text{s.t.} \quad X \in \{0,1\}^{n \times n}, \ X1_n = 1_n, \ X^\top 1_n = 1_n$$

A widely-used special case is the Koopmans-Beckmann's QAP, where the cost is determined by flow ($F$) and distance ($D$) matrices, formulated as:

$$\min_X \quad \text{trace}(F^\top X D X^\top),$$

$$\text{s.t.} \quad X \in \{0,1\}^{n \times n}, \ X1_n = 1_n, \ X^\top 1_n = 1_n$$

The Koopmans-Beckmann formulation is a special case of Lawler's QAP where $K = D^\top \otimes F$. Our method, PLMA, is designed specifically for this latter formulation.

**Ro-TS** (Taillard, 1991) is a highly optimized tabu search heuristic specifically for the Koopmans-Beckmann's QAP. We use the official C implementation from Éric Taillard's website[1].

**IPFP** (Leordeanu et al., 2009) is an iterative algorithm that seeks a good discrete solution by optimizing a continuous relaxation. While initially designed for Lawler's QAP, we use an improved version tailored for the Koopmans-Beckmann's QAP, as detailed in Algorithm 4. To ensure comparable computation time, we run IPFP with a batch of random initializations and report the best solution found.

---

**Algorithm 4** IPFP for Koopmans-Beckmann's QAP

---

**Require:** Initial permutation matrix $X_0$, number of max iterations $K$, convergence tolerance tol.
1: Initialize best solution $X^* \leftarrow X_0$.
2: **for** $k = 0, \dots, K-1$ **do**
3:     Calculate the gradient $\nabla f(X_k) = F X_k D^\top + F^\top X_k D$.
4:     Let $B_k \leftarrow \arg\min_{B \in \Pi_n} \langle B, \nabla f(X_k) \rangle$ via Hungarian algorithm.
5:     Let $a = \langle F, (B_k - X_k)D(B_k^\top - X_k^\top) \rangle$, $b = \langle F, (B_k - X_k)DX_k^\top + X_k D(B_k^\top - X_k^\top) \rangle$.
6:     **if** $a > 0$ **then**
7:         $t_k = \min(-\frac{b}{2a}, 1)$
8:     **else**
9:         $t_k = 1$
10:    **end if**
11:    Update $X_{k+1} \leftarrow X_k + t_k(B_k - X_k)$
12:    **if** $f(B_k) < f(X^*)$ **then**
13:        $X^* \leftarrow B_k$
14:    **end if**
15:    **if** $\|X_{k+1} - X_k\|_F \leq$ tol **then**
16:        **break**
17:    **end if**
18: **end for**
19: **return** The best found solution $X^*$.

---

**SM** (Leordeanu & Hebert, 2005) solves the Lawler's QAP by constructing an association graph and applying spectral methods to find the principal eigenvector, which is then discretized to obtain the final matching.

**RRWM** (Cho et al., 2010) also addresses the Lawler's QAP. It models the matching problem as a random walk on the association graph, introducing a re-weighting scheme to better preserve matching constraints during the walk.

**SAWT** (Tan & Mu, 2024) is a reinforcement learning method that learns to improve solutions for the Koopmans-Beckmann's QAP. A key distinction of SAWT is that it requires 2D coordinates for the locations as input, which limits its direct applicability to datasets where the distance matrix $D$ is not derived from Euclidean distances. Since the authors did not provide a canonical way to handle

---

[1]https://mistic.iict-heig-vd.ch/taillard/codes.dir/tabou_qap2.c

this, we adapted the implementation for the QAPLIB dataset by using Multi-dimensional Scaling (MDS) for symmetric matrices and Principal Component Analysis (PCA) for asymmetric matrices to generate the required 2D coordinates. For large-scale instances, we trained a new model with a larger initialization size ($N\_init = 512$) to avoid the limitations of the provided pre-trained models ($N\_init = 128$).

**NGM** (Wang et al., 2021b) approaches the problem by framing it as a vertex classification task on the association graph. We include NGM as a representative of learning-based methods that operate on the $n^2$ association graph, which, as noted in our main paper, can face scalability challenges.

**AR-Seq** uses an auto-regressive model and generates solutions by sequentially constructive sampling. Given a heatmap $\phi = \phi(\theta, \mathcal{P}) \in \mathbb{R}^{n \times n}$, our method constructs an energy-based model. As an alternative, one can define an auto-regressive model $q_\theta$ as follows

$$q_\theta(\pi) := \prod_{i=1}^{n} \frac{\exp(\phi_{i,\pi(i)})}{\sum_{j=i}^{n} \exp(\phi_{i,\pi(j)})}.$$

This factorization yields a sequence of conditional probabilities and therefore admits sequential constructive sampling. Owing to its ease of sampling, this formulation is widely used in prior active search methods.

## C  EXPERIMENTAL DETAILS

### C.1  MODEL ARCHITECTURE DETAILS

Our cross-graph attention network consists of a series of graph neural network layers followed by cross-attention blocks. The specific architectural parameters are listed in Table 5.

Table 5: Model architecture parameters for PLMA.

| Parameter | Value |
|---|---|
| Initial Feature Dimension ($d_{in}$) | 16 |
| Node Embedding Dimension ($d$) | 256 |
| GNN Layers ($l_1$) | 10 |
| Cross-Attention Blocks | 1 |
| Attention Heads per Block | 8 |
| Log-Sinkhorn Iterations | 1 |

### C.2  TRAINING AND FINETUNING

All models were trained using the PyTorch framework. The hyperparameters used for pre-training and warm-started MCMC finetuning are detailed in Table 6.

Table 6: Hyperparameters for PLMA.

| Parameter | Pre-training | Finetuning |
|---|---|---|
| Optimizer | Adam | Adam |
| Learning Rate | 1e-4 | 1e-4 |
| Batch Size | 64 | 256 |
| Training Steps | 468,900 | – |
| Total Finetuning Steps ($T$) | – | 200 |
| Starting Points ($K$) | 400 | 20 |
| Chains per Point ($M$) | - | 20 |
| Chain Length ($L$) | $n$ | $\lfloor n/3 \rfloor$ |
| Local Search Iterations ($T_{LS}$) | 1 | $n$ |

Table 7: Efficiency comparison on the uniformly random dataset with $n = 100$. PLMA (instance-wise) runs the same finetuning independently on each test instance without batching.

| Algorithm | Cost | Gap | Time |
|---|---|---|---|
| Ro-TS (1k) | 2195.98 | 0.13% | 38m59s |
| Ro-TS (5k) | 2193.16 | 0.00% | 3h15m |
| **PLMA (instance-wise)** | 2193.96 | 0.05% | 31m15s |
| **PLMA** | 2193.13 | 0.00% | 14m1s |

### C.3 HARDWARE

All experiments are conducted on a server with NVIDIA Tesla A800 GPUs (80GB) and Intel Xeon Gold 6326 CPUs (256GB) at 2.90GHz.

## D ADDITIONAL RESULTS

### D.1 EFFICIENCY OF BATCHED FINETUNING

In Table 7, we compare the efficiency the batch wise and instance wise variants of PLMA on the uniform random dataset with $n = 100$ and 200 finetuning steps. The batched configuration achieves a substantial reduction in runtime relative to the instance wise variant. Despite sharing one set of network parameters across instances during updates the batched variant preserves solution quality at the level of the strongest baseline. These observations support batched warm-started finetuning as an effective deployment-time adaptation strategy that improves computational efficiency without compromising final quality. Moreover, the instance-wise variant remains efficient in its own right and outperforms Ro-TS in runtime under comparable settings while delivering near-optimal solutions.

### D.2 HYPERPARAMETER STUDY

To analyze the sensitivity of our framework and validate its design, we conduct a hyperparameter study for the warm-started MCMC finetuning stage (Algorithm 1). This analysis was performed on the uniformly random dataset ($n = 100$) and the results are shown in Figure 3. We focused on three key hyperparameters governing the sampling process: the Markov chain length $L$, the number of initial starting points $K$, and the number of parallel chains per starting point $M$.

**Effect of Markov Chain Length ($L$).** Figure 3a shows the impact of varying $L$. The results confirm that a short-chain configuration is highly effective for adaptation. Optimal performance is achieved with $L = \lfloor n/3 \rfloor \approx 33$. This finding supports our core design principle: since the chains are warm-started from high-quality solutions found in the previous iteration, only a limited local exploration is needed for refinement. Excessively long chains risk straying from the promising region and wasting computation, while chains that are too short may not adequately explore the local neighborhood.

**Effect of ($K$, $M$) Configuration.** In Figure 3b, we study the balance between the diversity of starting points (controlled by $K$) and the search intensity around each point (controlled by $M$), keeping the total number of samples constant ($K \times M = 400$). The results show that extreme configurations perform poorly. For instance, using many starting points with minimal exploration each (($K, M) = (400, 1)$) lacks the search intensity to refine solutions, whereas concentrating all resources on a single starting point (($K, M) = (1, 400)$) suffers from a lack of initial diversity. The balanced configuration of $(K, M) = (20, 20)$ yields the best results, highlighting the benefit of exploring from a diverse set of promising solutions, with a sufficient search effort allocated to each.

### D.3 ITERATION CURVES ON SYNTHETIC DATASETS

Figure 4 displays the convergence behavior of PLMA during the adaptation phase by plotting the optimality gap against the number of finetuning steps ($T$). The curves reveal that PLMA's convergence dynamics differ notably across the two synthetic datasets, which may reflect a combination of the datasets' intrinsic properties and their interaction with our learning-based approach.

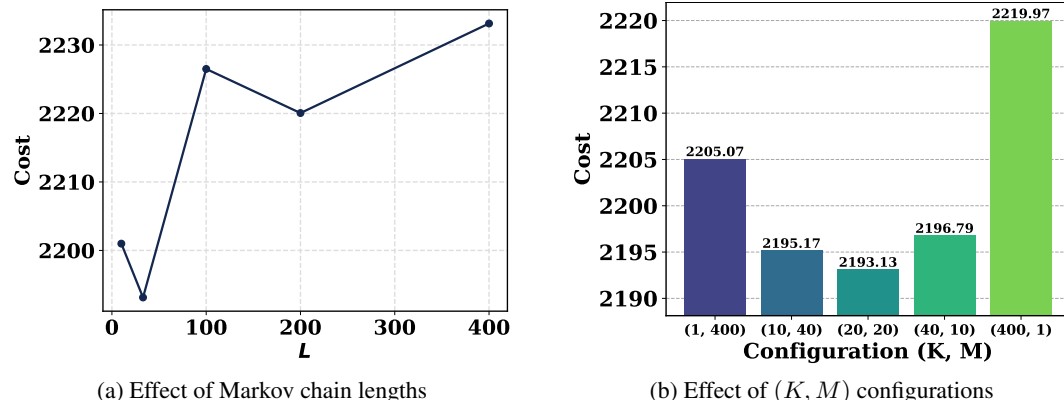

(a) Effect of Markov chain lengths

(b) Effect of $(K, M)$ configurations

Figure 3: Hyperparameter study

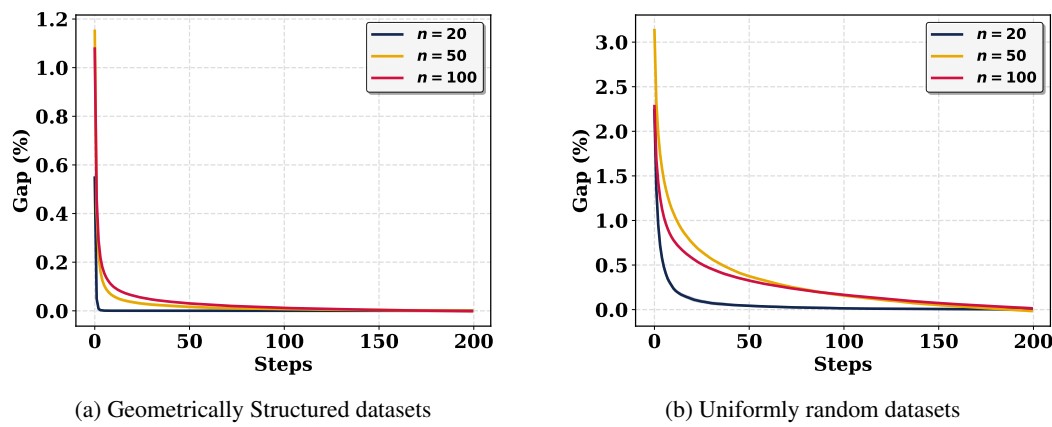

(a) Geometrically Structured datasets

(b) Uniformly random datasets

Figure 4: Iteration curves on synthetic datasets

On the geometrically structured instances, PLMA converges very quickly, with the optimality gap dropping sharply to near-zero within the initial finetuning steps. A plausible explanation is that the clear patterns in these instances are not only conducive to local search but are also effectively captured by our model during pre-training. This synergy may provide the adaptation phase with a highly advantageous starting point, enabling rapid convergence.

In contrast, on the uniformly random datasets, the convergence is more gradual but still steady. The lack of discernible patterns in these problems likely results in a more rugged and complex solution landscape. Consequently, the adaptation phase relies more heavily on iterative search to navigate the landscape, leading to a more extended convergence process. In either case, the results demonstrate the framework's robustness in consistently reaching high-quality solutions across varied problem structures.

## D.4 ANALYSIS OF OPTIMALITY GAP AND RUNTIME EFFICIENCY

To facilitate a clearer comparison of computational efficiency, we visualize the relationship between the optimality gap and wall-clock time in Figure 5. The results demonstrate that PLMA achieves superior solution quality for any given time budget compared to the baselines. Notably, on datasets with $n = 100$, PLMA achieves a zero optimality gap more than one order of magnitude faster than Ro-TS. This performance highlights the efficacy of our two-stage framework. The pre-trained model provides an instant, high-quality solution that outperforms several learning-based methods, while the subsequent MCMC fine-tuning rapidly refines this solution to near-optimality. These visualizations confirm that PLMA establishes a new state-of-the-art by offering the most favorable balance between optimality gap and runtime.

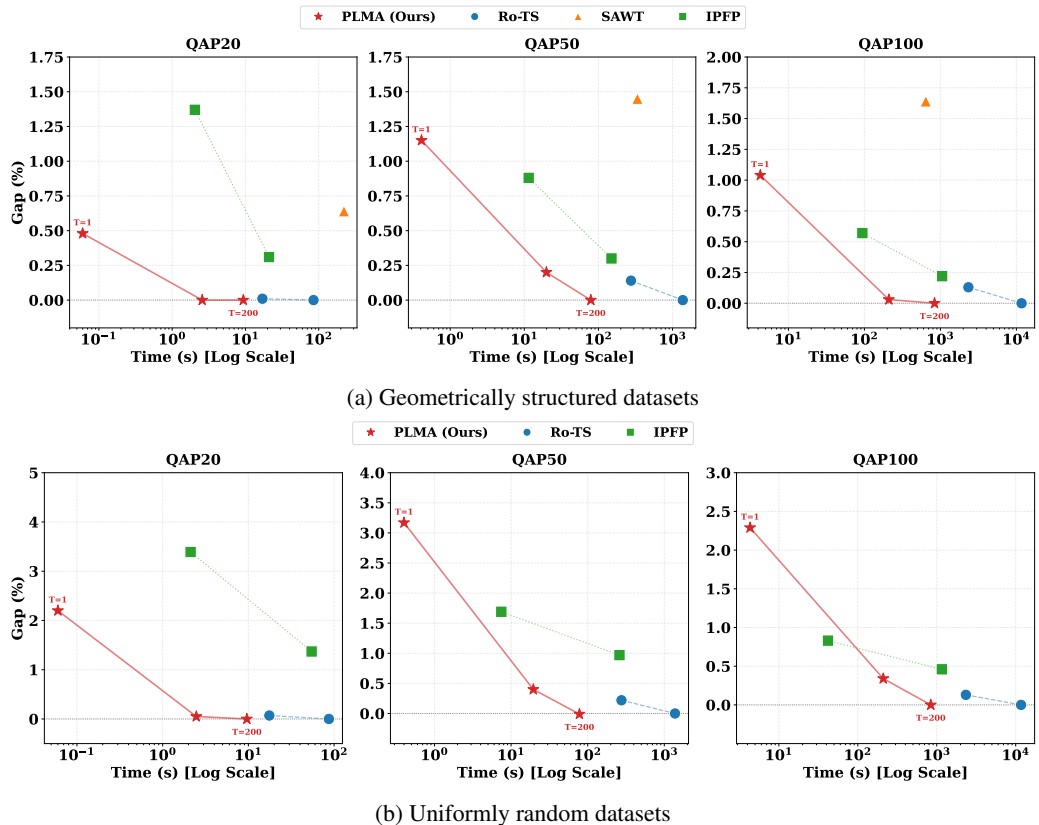

(a) Geometrically structured datasets

(b) Uniformly random datasets

Figure 5: Optimality gap vs. wall-clock time on synthetic datasets.

## D.5 IMPACT OF PRETRAINING

To assess the impact of pretraining, we compare the performance of finetuning a pretrained pol-
icy versus finetuning a randomly initialized policy on both the uniformly random dataset and the
Taixxeyy benchmark. The results are presented in Figure 6. The pretrained policy is trained on uni-
formly random instances with $n = 100$. Consequently, the uniformly random test dataset represents
in-distribution evaluation, while the Taixxeyy benchmark represents out-of-distribution evaluation.

As shown in Figure 6a, the pretrained policy achieves a much lower optimality gap faster. This
confirms that the transferable knowledge acquired during pretraining provides a powerful head start
by guiding the policy's updates toward more optimal actions. Indeed, given a sufficient number
of finetuning steps, the random policy can eventually achieve results comparable to the pretrained
policy. This convergence speaks to the effectiveness of our underlying warm-started MCMC fine-
tuning, which is powerful enough to eventually rediscover high-quality solutions from a random
start. However, the transferable knowledge from pretraining provides a distinct and critical advan-
tage in finetuning efficiency, enabling the policy to provide superior solutions significantly faster in
the early, more practical stages of finetuning.

The importance of pretraining is far more pronounced on the notably difficult Taixxeyy instances.
As shown in Figure 6b, the pretrained policy consistently and significantly outperforms the random
policy. This stark performance gap demonstrates that the pretraining on synthetic data endows the
policy with meaningful and general knowledge about the fundamental structure of the QAP. Lacking
these general-purpose structural priors, the random policy struggles to adapt to the unique complex-
ities of Taixxeyy instances and is unable to converge to high-quality solutions, even with a large
number of finetuning steps.

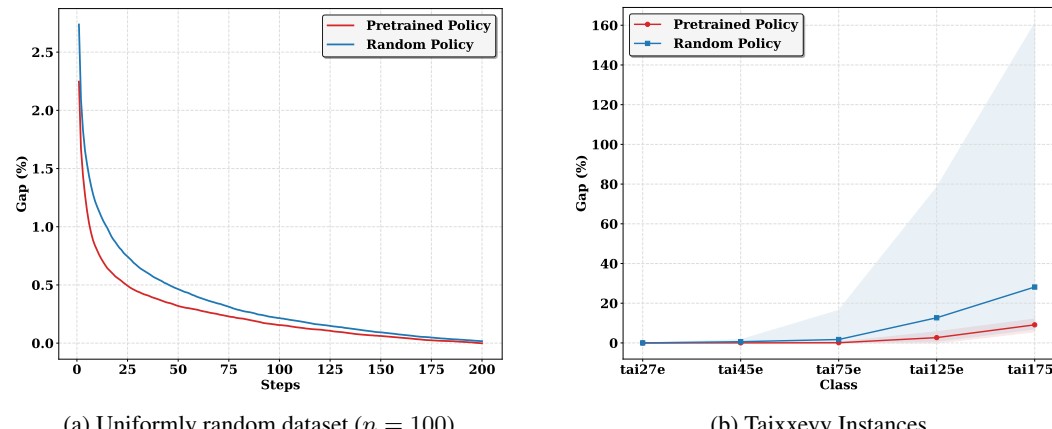

(a) Uniformly random dataset ($n = 100$)   (b) Taixxeyy Instances

Figure 6: Performance comparison between finetuning from a pretrained policy and finetuning from a randomly initialized policy. **Left:** Iteration curves on the uniformly random dataset ($n = 100$). **Right:** Comparative analysis on the challenging Taixxeyy instances across different problem classes. The solid lines denote the mean optimality gap, while the shaded regions encapsulate the interval between the minimum and maximum gaps observed across 10 independent trials.

### D.6 DETAILED RESULTS ON QAPLIB

For a comprehensive evaluation, we present a detailed breakdown of our model's performance on each instance of the QAPLIB benchmark in Table 8. To account for the stochastic nature of the solver, each algorithm was executed for 10 independent runs. The table lists the minimum, mean, and maximum gaps and average computation time across the 10 runs.

### D.7 DETAILED RESULTS ON TAIXXEYY

We also present the detailed results for the challenging Taixxeyy datasets. As noted by Drezner et al. (2005), in these instances, the variance of the solutions for these instances can be very high, making it injudicious to rely solely on the average and standard deviation of solution values. In Table 9, we presents the minimum, mean, and maximum gaps and average computation time across the 10 runs for each method.

Table 8: Detailed comparison on QAPLIB instances (10 repetitions). Each cell shows (min/mean/max) Gap (%) and average Time (s).

| Problem | Ro-TS | IPFP | SAWT | PLMA |
|---|---|---|---|---|
| **Name** | Gap / Time | Gap / Time | Gap / Time | Gap / Time |
| bur26a | (0.00, **0.00**, 0.00) / 0.08 | (0.04, 0.10, 0.14) / 0.38 | (2.81, 3.74, 4.45) / 14.77 | (0.00, **0.00**, 0.00) / 0.18 |
| bur26b | (0.00, **0.00**, 0.00) / 0.11 | (0.01, 0.06, 0.23) / 0.40 | (3.27, 3.81, 4.63) / 13.90 | (0.00, **0.00**, 0.00) / 0.04 |
| bur26c | (0.00, **0.00**, 0.00) / 0.10 | (0.01, 0.03, 0.04) / 0.37 | (2.43, 4.08, 4.83) / 13.89 | (0.00, **0.00**, 0.00) / 0.04 |
| bur26d | (0.00, **0.00**, 0.00) / 0.12 | (0.01, 0.03, 0.05) / 0.40 | (3.78, 4.41, 5.08) / 13.87 | (0.00, 0.00, 0.00) / 0.06 |
| bur26e | (0.00, **0.00**, 0.00) / 0.08 | (0.01, 0.02, 0.04) / 0.23 | (2.73, 4.37, 5.48) / 13.90 | (0.00, **0.00**, 0.00) / 0.08 |
| bur26f | (0.00, **0.00**, 0.00) / 0.10 | (0.02, 0.02, 0.03) / 0.33 | (4.03, 4.96, 6.21) / 13.89 | (0.00, **0.00**, 0.00) / 0.05 |
| bur26g | (0.00, **0.00**, 0.00) / 0.11 | (0.04, 0.06, 0.10) / 0.25 | (2.95, 3.63, 4.55) / 13.88 | (0.00, **0.00**, 0.00) / 0.06 |
| bur26h | (0.00, **0.00**, 0.00) / 0.09 | (0.02, 0.04, 0.06) / 0.25 | (2.62, 3.91, 4.90) / 13.88 | (0.00, **0.00**, 0.00) / 0.07 |
| chr12a | (0.00, **0.00**, 0.00) / 0.00 | (3.81, 8.62, 18.40) / 0.21 | (45.35, 88.99, 116.21) / 13.32 | (0.00, **0.00**, 0.00) / 0.02 |
| chr12b | (0.00, **0.00**, 0.00) / 0.00 | (0.00, **0.00**, 0.00) / 0.19 | (43.73, 88.49, 126.24) / 13.32 | (0.00, **0.00**, 0.00) / 0.01 |
| chr12c | (0.00, **0.00**, 0.00) / 0.01 | (3.68, 12.58, 19.90) / 0.23 | (29.69, 67.48, 89.71) / 13.23 | (0.00, 0.03, 0.27) / 0.90 |
| chr15a | (0.00, **0.00**, 0.00) / 0.03 | (4.81, 16.94, 37.45) / 0.25 | (121.79, 158.68, 188.20) / 13.29 | (0.00, **0.00**, 0.00) / 0.32 |
| chr15b | (0.00, **0.00**, 0.00) / 0.02 | (7.96, 13.81, 20.35) / 0.16 | (142.35, 198.93, 251.41) / 13.20 | (0.00, 0.80, 7.96) / 1.55 |
| chr15c | (0.00, **0.00**, 0.00) / 0.03 | (16.06, 29.30, 41.37) / 0.26 | (127.27, 182.16, 229.00) / 13.22 | (0.00, **0.00**, 0.00) / 0.14 |
| chr18a | (0.00, **0.00**, 0.00) / 0.02 | (0.18, 21.75, 30.10) / 0.23 | (208.27, 247.32, 288.23) / 13.48 | (0.00, **0.00**, 0.00) / 0.38 |
| chr18b | (0.00, **0.00**, 0.00) / 0.00 | (1.17, 4.45, 10.17) / 0.31 | (74.97, 86.31, 90.74) / 13.42 | (0.00, **0.00**, 0.00) / 0.04 |

| Problem | Ro-TS | IPFP | SAWT | PLMA |
|---|---|---|---|---|
| Name | Gap / Time | Gap / Time | Gap / Time | Gap / Time |
| chr20a | (0.00, 0.74, 4.38) / 0.21 | (6.39, 17.01, 23.45) / 0.24 | (137.04, 170.41, 198.91) / 13.47 | (0.00, **0.17**, 1.37) / 4.46 |
| chr20b | (0.00, 2.72, 4.53) / 0.33 | (11.05, 16.18, 19.41) / 0.27 | (156.31, 177.75, 187.12) / 13.49 | (0.00, **1.86**, 3.66) / 4.31 |
| chr20c | (0.00, **0.00**, 0.00) / 0.10 | (0.00, 12.02, 21.37) / 0.13 | (164.32, 258.89, 322.98) / 13.50 | (0.00, **0.00**, 0.00) / 0.29 |
| chr22a | (0.00, 0.20, 0.75) / 0.36 | (9.81, 12.36, 14.81) / 0.40 | (44.31, 56.17, 67.90) / 13.48 | (0.00, **0.00**, 0.00) / 0.18 |
| chr22b | (0.00, 0.47, 1.68) / 0.27 | (12.98, 15.00, 17.79) / 0.38 | (46.04, 53.95, 60.80) / 13.46 | (0.00, **0.00**, 0.00) / 0.40 |
| chr25a | (0.00, 3.21, 6.74) / 0.57 | (18.07, 28.65, 44.47) / 0.39 | (198.16, 242.87, 266.49) / 13.55 | (0.00, **0.00**, 0.00) / 0.22 |
| els19 | (0.00, **0.00**, 0.00) / 0.01 | (1.44, 10.18, 33.73) / 0.36 | (47.37, 47.37, 47.37) / 13.37 | (0.00, **0.00**, 0.00) / 0.08 |
| esc128 | (0.00, **0.00**, 0.00) / 9.04 | (0.00, **0.00**, 0.00) / 4.63 | (125.00, 182.81, 206.25) / 21.46 | (0.00, **0.00**, 0.00) / 0.10 |
| esc16a | (0.00, **0.00**, 0.00) / 0.00 | (0.00, **0.00**, 0.00) / 0.29 | (5.88, 17.06, 26.47) / 13.11 | (0.00, **0.00**, 0.00) / 0.02 |
| esc16b | (0.00, **0.00**, 0.00) / 0.00 | (0.00, **0.00**, 0.00) / 0.37 | (0.00, 0.27, 0.68) / 13.10 | (0.00, **0.00**, 0.00) / 0.02 |
| esc16c | (0.00, **0.00**, 0.00) / 0.00 | (0.00, **0.00**, 0.00) / 0.37 | (5.00, 14.50, 21.25) / 13.11 | (0.00, **0.00**, 0.00) / 0.02 |
| esc16d | (0.00, **0.00**, 0.00) / 0.00 | (0.00, **0.00**, 0.00) / 0.28 | (25.00, 43.75, 62.50) / 13.10 | (0.00, **0.00**, 0.00) / 0.02 |
| esc16e | (0.00, **0.00**, 0.00) / 0.00 | (0.00, **0.00**, 0.00) / 0.24 | (14.29, 17.14, 21.43) / 13.09 | (0.00, **0.00**, 0.00) / 0.02 |
| esc16g | (0.00, **0.00**, 0.00) / 0.00 | (0.00, **0.00**, 0.00) / 0.19 | (7.69, 14.62, 15.38) / 13.08 | (0.00, **0.00**, 0.00) / 0.02 |
| esc16h | (0.00, **0.00**, 0.00) / 0.00 | (0.00, **0.00**, 0.00) / 0.24 | (1.61, 6.12, 8.84) / 13.10 | (0.00, **0.00**, 0.00) / 0.02 |
| esc16i | (0.00, **0.00**, 0.00) / 0.00 | (0.00, **0.00**, 0.00) / 0.19 | (28.57, 65.71, 114.29) / 13.09 | (0.00, **0.00**, 0.00) / 0.02 |
| esc16j | (0.00, **0.00**, 0.00) / 0.00 | (0.00, **0.00**, 0.00) / 0.09 | (0.00, 37.50, 50.00) / 13.08 | (0.00, **0.00**, 0.00) / 0.02 |
| esc32a | (0.00, **0.00**, 0.00) / 0.39 | (3.08, 5.38, 7.69) / 0.42 | (138.46, 157.08, 173.85) / 13.74 | (0.00, **0.00**, 0.00) / 0.27 |
| esc32b | (0.00, **0.00**, 0.00) / 0.05 | (0.00, 0.95, 9.52) / 0.22 | (90.48, 90.48, 90.48) / 13.71 | (0.00, **0.00**, 0.00) / 0.08 |
| esc32c | (0.00, **0.00**, 0.00) / 0.00 | (0.00, **0.00**, 0.00) / 0.54 | (7.17, 10.09, 12.15) / 13.73 | (0.00, **0.00**, 0.00) / 0.03 |
| esc32d | (0.00, **0.00**, 0.00) / 0.03 | (0.00, **0.00**, 0.00) / 0.41 | (28.00, 33.50, 43.00) / 13.73 | (0.00, **0.00**, 0.00) / 0.03 |
| esc32e | (0.00, **0.00**, 0.00) / 0.00 | (0.00, **0.00**, 0.00) / 0.30 | (0.00, **0.00**, 0.00) / 13.73 | (0.00, **0.00**, 0.00) / 0.03 |
| esc32g | (0.00, **0.00**, 0.00) / 0.00 | (0.00, **0.00**, 0.00) / 0.25 | (0.00, 16.67, 66.67) / 13.72 | (0.00, **0.00**, 0.00) / 0.03 |
| esc32h | (0.00, **0.00**, 0.00) / 0.04 | (0.46, 0.73, 0.91) / 0.60 | (19.18, 21.87, 26.03) / 13.76 | (0.00, **0.00**, 0.00) / 0.03 |
| esc64a | (0.00, **0.00**, 0.00) / 0.19 | (0.00, **0.00**, 0.00) / 1.69 | (43.10, 74.14, 101.72) / 19.00 | (0.00, **0.00**, 0.00) / 0.04 |
| had12 | (0.00, **0.00**, 0.00) / 0.00 | (0.24, 0.27, 0.36) / 0.33 | (3.51, 4.78, 6.17) / 13.02 | (0.00, **0.00**, 0.00) / 0.02 |
| had14 | (0.00, **0.00**, 0.00) / 0.00 | (0.00, **0.00**, 0.00) / 0.33 | (3.16, 5.90, 7.56) / 13.07 | (0.00, **0.00**, 0.00) / 0.02 |
| had16 | (0.00, **0.00**, 0.00) / 0.01 | (0.05, 0.06, 0.11) / 0.37 | (3.82, 5.61, 6.51) / 13.12 | (0.00, **0.00**, 0.00) / 0.02 |
| had18 | (0.00, **0.00**, 0.00) / 0.01 | (0.00, 0.01, 0.04) / 0.46 | (4.07, 5.16, 6.20) / 13.36 | (0.00, **0.00**, 0.00) / 0.02 |
| had20 | (0.00, **0.00**, 0.00) / 0.01 | (0.03, 0.05, 0.06) / 0.45 | (4.13, 5.44, 7.22) / 13.44 | (0.00, **0.00**, 0.00) / 0.02 |
| kra30a | (0.00, **0.00**, 0.00) / 0.24 | (0.00, 0.62, 1.77) / 0.65 | (27.55, 33.24, 36.31) / 13.66 | (0.00, **0.00**, 0.00) / 0.78 |
| kra30b | (0.00, **0.00**, 0.00) / 0.20 | (0.00, 0.21, 0.72) / 0.64 | (27.02, 32.08, 35.05) / 13.69 | (0.00, **0.00**, 0.00) / 1.41 |
| kra32 | (0.00, **0.00**, 0.00) / 0.10 | (0.00, 1.12, 1.76) / 0.57 | (31.87, 34.77, 36.83) / 13.74 | (0.00, **0.00**, 0.00) / 0.11 |
| lipa20a | (0.00, **0.00**, 0.00) / 0.01 | (0.00, 0.70, 2.20) / 0.71 | (4.51, 4.90, 5.21) / 13.39 | (0.00, **0.00**, 0.00) / 0.05 |
| lipa20b | (0.00, **0.00**, 0.00) / 0.00 | (0.00, **0.00**, 0.00) / 0.36 | (0.00, **0.00**, 0.00) / 13.41 | (0.00, **0.00**, 0.00) / 0.02 |
| lipa30a | (0.00, **0.00**, 0.00) / 0.03 | (0.18, 1.69, 2.00) / 1.27 | (3.56, 3.72, 3.86) / 13.69 | (0.00, **0.00**, 0.00) / 0.12 |
| lipa30b | (0.00, **0.00**, 0.00) / 0.01 | (0.00, **0.00**, 0.00) / 0.69 | (0.00, **0.00**, 0.00) / 13.68 | (0.00, **0.00**, 0.00) / 0.05 |
| lipa40a | (0.00, **0.00**, 0.00) / 0.10 | (1.53, 1.59, 1.66) / 1.99 | (2.90, 3.02, 3.11) / 13.98 | (0.00, **0.00**, 0.00) / 0.25 |
| lipa40b | (0.00, **0.00**, 0.00) / 0.01 | (0.00, **0.00**, 0.00) / 1.00 | (0.00, **0.00**, 0.00) / 13.99 | (0.00, **0.00**, 0.00) / 0.06 |
| lipa50a | (0.00, **0.00**, 0.00) / 0.21 | (1.36, 1.41, 1.44) / 4.32 | (2.40, 2.62, 2.72) / 14.25 | (0.00, **0.00**, 0.00) / 0.64 |
| lipa50b | (0.00, **0.00**, 0.00) / 0.05 | (0.00, **0.00**, 0.00) / 2.33 | (0.00, **0.00**, 0.00) / 14.28 | (0.00, **0.00**, 0.00) / 0.07 |
| lipa60a | (0.00, **0.00**, 0.00) / 0.80 | (1.11, 1.18, 1.23) / 6.26 | (2.15, 2.29, 2.35) / 18.78 | (0.00, 0.06, 0.55) / 3.66 |
| lipa60b | (0.00, **0.00**, 0.00) / 0.11 | (0.00, **0.00**, 0.00) / 3.34 | (0.00, **0.00**, 0.00) / 18.75 | (0.00, **0.00**, 0.00) / 0.11 |
| lipa70a | (0.00, **0.00**, 0.00) / 2.78 | (1.06, 1.07, 1.11) / 8.87 | (2.04, 2.08, 2.12) / 19.19 | (0.00, 0.14, 0.76) / 4.70 |
| lipa70b | (0.00, **0.00**, 0.00) / 0.28 | (0.00, 3.74, 18.72) / 4.73 | (0.00, **0.00**, 0.00) / 19.17 | (0.00, **0.00**, 0.00) / 0.12 |
| lipa80a | (0.00, 0.10, 0.53) / 10.14 | (0.95, 0.98, 1.01) / 12.77 | (1.79, 1.85, 1.90) / 19.57 | (0.67, 0.71, 0.73) / 9.45 |
| lipa80b | (0.00, **0.00**, 0.00) / 0.66 | (0.00, 1.97, 19.69) / 5.93 | (0.00, **0.00**, 0.00) / 19.57 | (0.00, **0.00**, 0.00) / 0.22 |
| lipa90a | (0.00, 0.19, 0.48) / 21.83 | (0.86, 0.89, 0.92) / 16.70 | (1.64, 1.68, 1.72) / 19.97 | (0.64, 0.66, 0.68) / 8.47 |
| lipa90b | (0.00, **0.00**, 0.00) / 0.51 | (0.00, 1.99, 19.90) / 7.68 | (0.00, **0.00**, 0.00) / 19.95 | (0.00, **0.00**, 0.00) / 0.24 |
| nug12 | (0.00, **0.00**, 0.00) / 0.00 | (0.00, **0.00**, 0.00) / 0.24 | (10.73, 15.47, 18.34) / 13.01 | (0.00, **0.00**, 0.00) / 0.01 |
| nug14 | (0.00, **0.00**, 0.00) / 0.01 | (0.00, 0.02, 0.20) / 0.30 | (11.44, 14.00, 15.58) / 13.07 | (0.00, **0.00**, 0.00) / 0.03 |
| nug15 | (0.00, **0.00**, 0.00) / 0.00 | (0.00, **0.00**, 0.00) / 0.31 | (12.52, 16.02, 19.48) / 13.07 | (0.00, **0.00**, 0.00) / 0.02 |
| nug16a | (0.00, **0.00**, 0.00) / 0.00 | (0.00, 0.09, 0.75) / 0.30 | (9.81, 15.24, 18.63) / 13.09 | (0.00, **0.00**, 0.00) / 0.04 |
| nug16b | (0.00, **0.00**, 0.00) / 0.00 | (0.00, **0.00**, 0.00) / 0.30 | (13.71, 18.50, 21.94) / 13.11 | (0.00, **0.00**, 0.00) / 0.02 |
| nug17 | (0.00, **0.00**, 0.00) / 0.02 | (0.00, 0.02, 0.12) / 0.33 | (13.63, 16.57, 19.52) / 13.30 | (0.00, **0.00**, 0.00) / 0.06 |
| nug18 | (0.00, **0.00**, 0.00) / 0.01 | (0.00, 0.04, 0.41) / 0.33 | (13.68, 16.46, 19.79) / 13.37 | (0.00, **0.00**, 0.00) / 0.25 |
| nug20 | (0.00, **0.00**, 0.00) / 0.01 | (0.00, 0.03, 0.31) / 0.37 | (14.24, 16.86, 19.22) / 13.42 | (0.00, **0.00**, 0.00) / 0.05 |

| Problem | Ro-TS | IPFP | SAWT | PLMA |
|---------|-------|------|------|------|
| **Name** | Gap / Time | Gap / Time | Gap / Time | Gap / Time |
| nug21 | (0.00, **0.00**, 0.00) / 0.02 | (0.00, 0.19, 0.41) / 0.39 | (19.85, 22.46, 25.43) / 13.42 | (0.00, **0.00**, 0.00) / 0.06 |
| nug22 | (0.00, **0.00**, 0.00) / 0.01 | (0.00, **0.00**, 0.00) / 0.38 | (18.35, 21.69, 24.25) / 13.49 | (0.00, **0.00**, 0.00) / 0.05 |
| nug24 | (0.00, **0.00**, 0.00) / 0.01 | (0.00, 0.01, 0.11) / 0.43 | (18.92, 22.00, 24.43) / 13.50 | (0.00, **0.00**, 0.00) / 0.09 |
| nug25 | (0.00, **0.00**, 0.00) / 0.01 | (0.11, 0.11, 0.16) / 0.47 | (17.84, 19.49, 23.24) / 13.52 | (0.00, **0.00**, 0.00) / 0.10 |
| nug27 | (0.00, **0.00**, 0.00) / 0.02 | (0.04, 0.04, 0.04) / 0.51 | (20.10, 22.20, 24.00) / 13.58 | (0.00, **0.00**, 0.00) / 0.08 |
| nug28 | (0.00, **0.00**, 0.00) / 0.06 | (0.00, 0.15, 0.39) / 0.52 | (19.09, 22.02, 23.50) / 13.60 | (0.00, **0.00**, 0.00) / 0.12 |
| nug30 | (0.00, **0.00**, 0.00) / 0.17 | (0.00, 0.04, 0.07) / 0.60 | (20.08, 21.86, 23.74) / 13.69 | (0.00, 0.01, 0.07) / 1.19 |
| rou12 | (0.00, **0.00**, 0.00) / 0.00 | (0.00, 0.38, 1.45) / 0.29 | (9.88, 12.06, 14.99) / 13.00 | (0.00, **0.00**, 0.00) / 0.01 |
| rou15 | (0.00, **0.00**, 0.00) / 0.01 | (0.00, 1.20, 3.19) / 0.30 | (12.25, 16.09, 18.39) / 13.09 | (0.00, **0.00**, 0.00) / 0.05 |
| rou20 | (0.00, **0.00**, 0.00) / 0.12 | (0.40, 0.73, 1.16) / 0.39 | (12.21, 14.94, 16.64) / 13.40 | (0.00, 0.05, 0.18) / 5.25 |
| scr12 | (0.00, **0.00**, 0.00) / 0.00 | (0.00, 1.35, 3.87) / 0.23 | (15.47, 19.07, 26.54) / 12.98 | (0.00, **0.00**, 0.00) / 0.01 |
| scr15 | (0.00, **0.00**, 0.00) / 0.00 | (0.00, 0.94, 4.12) / 0.26 | (30.38, 34.71, 42.59) / 13.06 | (0.00, **0.00**, 0.00) / 0.03 |
| scr20 | (0.00, **0.00**, 0.00) / 0.01 | (0.03, 1.44, 2.85) / 0.34 | (38.23, 50.37, 56.50) / 13.37 | (0.00, **0.00**, 0.00) / 0.08 |
| sko100a | (0.04, 0.08, 0.11) / 45.80 | (0.22, 0.32, 0.42) / 12.58 | (13.96, 14.71, 15.64) / 19.51 | (0.02, **0.06**, 0.11) / 9.39 |
| sko100b | (0.03, **0.04**, 0.07) / 45.79 | (0.12, 0.33, 0.53) / 12.68 | (13.59, 14.54, 15.12) / 19.44 | (0.01, 0.05, 0.11) / 9.37 |
| sko100c | (0.02, 0.04, 0.06) / 45.77 | (0.09, 0.24, 0.52) / 13.04 | (14.43, 15.28, 15.58) / 19.42 | (0.00, **0.02**, 0.04) / 9.30 |
| sko100d | (0.03, 0.08, 0.13) / 45.73 | (0.20, 0.33, 0.45) / 12.94 | (13.99, 14.69, 15.24) / 19.44 | (0.03, **0.05**, 0.09) / 9.01 |
| sko100e | (0.01, 0.04, 0.05) / 45.81 | (0.12, 0.40, 0.56) / 12.85 | (14.34, 15.40, 15.89) / 19.37 | (0.01, **0.03**, 0.05) / 9.12 |
| sko100f | (0.05, 0.09, 0.15) / 45.80 | (0.39, 0.47, 0.56) / 12.80 | (13.57, 14.33, 14.76) / 19.37 | (0.01, **0.07**, 0.16) / 7.40 |
| sko42 | (0.00, **0.00**, 0.00) / 1.08 | (0.04, 0.18, 0.35) / 1.55 | (18.52, 19.57, 20.72) / 13.99 | (0.00, **0.00**, 0.00) / 0.40 |
| sko49 | (0.00, 0.04, 0.08) / 4.27 | (0.07, 0.27, 0.56) / 2.10 | (17.15, 18.45, 19.86) / 14.19 | (0.00, **0.03**, 0.07) / 3.75 |
| sko56 | (0.00, 0.02, 0.04) / 7.12 | (0.09, 0.29, 0.57) / 2.72 | (17.12, 18.27, 19.33) / 17.71 | (0.00, **0.01**, 0.02) / 4.83 |
| sko64 | (0.00, 0.01, 0.01) / 8.75 | (0.02, 0.23, 0.39) / 3.72 | (16.03, 16.90, 18.06) / 17.96 | (0.00, **0.00**, 0.01) / 3.99 |
| sko72 | (0.00, 0.05, 0.10) / 16.09 | (0.17, 0.28, 0.55) / 5.08 | (16.04, 16.46, 17.28) / 18.30 | (0.00, **0.03**, 0.09) / 8.27 |
| sko81 | (0.03, 0.06, 0.10) / 23.51 | (0.20, 0.31, 0.42) / 7.57 | (15.28, 15.72, 16.17) / 18.63 | (0.00, **0.03**, 0.08) / 8.43 |
| sko90 | (0.03, 0.06, 0.11) / 32.35 | (0.21, 0.31, 0.42) / 9.59 | (14.74, 15.19, 15.54) / 18.98 | (0.02, **0.06**, 0.10) / 9.06 |
| ste36a | (0.00, **0.00**, 0.00) / 0.78 | (0.82, 1.88, 3.99) / 0.75 | (56.02, 63.33, 64.52) / 13.83 | (0.00, 0.03, 0.25) / 1.59 |
| ste36b | (0.00, **0.00**, 0.00) / 0.18 | (0.56, 1.51, 3.13) / 0.48 | (172.62, 190.62, 206.91) / 13.83 | (0.00, **0.00**, 0.00) / 0.20 |
| ste36c | (0.00, **0.00**, 0.00) / 0.30 | (0.71, 2.05, 3.15) / 0.86 | (51.21, 62.17, 68.23) / 13.84 | (0.00, **0.00**, 0.00) / 0.50 |
| tai100a | (0.80, **0.97**, 1.11) / 45.80 | (1.22, 1.44, 1.56) / 8.22 | (12.27, 12.66, 12.87) / 19.45 | (0.83, 1.00, 1.25) / 6.57 |
| tai100b | (0.05, 0.18, 0.28) / 45.76 | (0.25, 0.64, 0.84) / 12.24 | (35.90, 38.97, 41.36) / 19.41 | (0.00, **0.01**, 0.02) / 6.68 |
| tai10a | (0.00, **0.00**, 0.00) / 0.00 | (0.00, 0.24, 0.59) / 0.24 | (2.30, 10.69, 16.82) / 12.90 | (0.00, **0.00**, 0.00) / 0.01 |
| tai12a | (0.00, **0.00**, 0.00) / 0.00 | (0.00, **0.00**, 0.00) / 0.27 | (10.83, 14.80, 18.07) / 12.99 | (0.00, **0.00**, 0.00) / 0.01 |
| tai12b | (0.00, **0.00**, 0.00) / 0.00 | (0.00, 0.32, 1.52) / 0.20 | (11.78, 21.86, 34.97) / 13.01 | (0.00, **0.00**, 0.00) / 0.01 |
| tai150b | (0.20, 0.39, 0.58) / 110.54 | (1.12, 1.29, 1.62) / 32.63 | (25.47, 26.25, 26.69) / 21.34 | (0.10, **0.26**, 0.39) / 12.75 |
| tai15a | (0.00, **0.00**, 0.00) / 0.01 | (0.20, 0.60, 1.04) / 0.33 | (9.74, 11.41, 13.86) / 13.05 | (0.00, **0.00**, 0.00) / 0.03 |
| tai15b | (0.00, **0.00**, 0.00) / 0.00 | (0.04, 0.20, 0.32) / 0.31 | (1.71, 2.39, 2.81) / 13.07 | (0.00, **0.00**, 0.00) / 0.02 |
| tai17a | (0.00, **0.00**, 0.00) / 0.02 | (0.54, 1.28, 2.07) / 0.37 | (12.79, 15.02, 16.55) / 13.30 | (0.00, **0.00**, 0.00) / 0.40 |
| tai20a | (0.00, **0.09**, 0.30) / 0.20 | (0.00, 0.91, 1.82) / 0.37 | (14.75, 16.50, 18.31) / 13.38 | (0.00, 0.17, 0.47) / 4.60 |
| tai20b | (0.00, **0.00**, 0.00) / 0.01 | (0.00, 0.31, 0.76) / 0.32 | (16.24, 34.75, 51.28) / 13.38 | (0.00, **0.00**, 0.00) / 0.05 |
| tai256c | (0.18, 0.21, 0.24) / 368.55 | (0.67, 0.80, 1.00) / 9.69 | (62.32, 90.52, 115.80) / 25.31 | (0.16, **0.18**, 0.22) / 36.74 |
| tai25a | (0.00, **0.04**, 0.37) / 0.34 | (0.88, 1.63, 2.32) / 0.52 | (15.02, 15.97, 16.91) / 13.52 | (0.37, 0.57, 0.78) / 8.34 |
| tai25b | (0.00, **0.00**, 0.00) / 0.06 | (0.09, 0.65, 1.39) / 0.37 | (36.75, 56.55, 71.64) / 13.53 | (0.00, **0.00**, 0.00) / 0.09 |
| tai30a | (0.00, **0.04**, 0.40) / 0.83 | (0.95, 1.47, 1.69) / 0.66 | (14.18, 14.76, 15.27) / 13.68 | (0.00, 0.65, 1.07) / 6.53 |
| tai30b | (0.00, 0.00, 0.00) / 0.56 | (0.00, 0.56, 2.22) / 0.54 | (44.16, 52.71, 62.29) / 13.67 | (0.00, **0.00**, 0.00) / 0.63 |
| tai35a | (0.00, **0.36**, 0.78) / 1.66 | (1.36, 1.72, 2.20) / 0.80 | (14.15, 15.26, 15.89) / 13.80 | (0.23, 0.63, 0.98) / 8.38 |
| tai35b | (0.00, **0.00**, 0.00) / 0.81 | (0.14, 0.66, 1.41) / 0.52 | (27.12, 43.65, 49.07) / 13.82 | (0.00, **0.00**, 0.00) / 0.71 |
| tai40a | (0.32, **0.53**, 0.68) / 2.76 | (0.98, 1.55, 2.00) / 0.86 | (13.72, 15.58, 16.24) / 13.96 | (0.33, 0.86, 1.27) / 7.16 |
| tai40b | (0.00, 0.00, 0.01) / 0.73 | (0.04, 0.46, 2.71) / 0.80 | (42.25, 46.83, 55.28) / 13.96 | (0.00, **0.00**, 0.00) / 0.39 |
| tai50a | (0.34, **0.81**, 1.20) / 5.41 | (1.22, 1.63, 1.91) / 1.82 | (14.32, 15.64, 16.71) / 14.22 | (0.67, 1.06, 1.37) / 5.46 |
| tai50b | (0.00, 0.11, 0.37) / 5.06 | (0.20, 0.80, 1.74) / 1.92 | (43.10, 45.39, 49.12) / 14.22 | (0.00, **0.00**, 0.00) / 1.64 |
| tai60a | (0.72, **0.99**, 1.18) / 9.38 | (1.41, 1.75, 2.11) / 2.58 | (14.67, 15.11, 15.42) / 17.97 | (0.95, 1.13, 1.40) / 6.60 |
| tai60b | (0.00, 0.02, 0.04) / 8.05 | (0.11, 0.43, 0.68) / 2.73 | (41.85, 47.35, 51.30) / 17.94 | (0.00, **0.00**, 0.00) / 2.28 |
| tai64c | (0.00, **0.00**, 0.00) / 0.54 | (0.24, 0.58, 1.15) / 0.21 | (217.55, 217.55, 217.55) / 18.08 | (0.00, **0.00**, 0.00) / 0.04 |
| tai80a | (0.95, 1.15, 1.28) / 22.67 | (1.44, 1.64, 1.86) / 4.40 | (13.27, 13.60, 13.88) / 18.74 | (0.83, **1.04**, 1.25) / 7.94 |
| tai80b | (0.00, 0.20, 0.66) / 22.61 | (0.42, 1.26, 1.92) / 6.02 | (37.65, 39.91, 42.42) / 18.73 | (0.00, **0.06**, 0.31) / 8.99 |
| tho150 | (0.06, **0.11**, 0.16) / 110.38 | (0.33, 0.51, 0.76) / 36.00 | (17.67, 18.09, 18.69) / 21.34 | (0.20, 0.26, 0.36) / 10.26 |

| Problem | Ro-TS | IPFP | SAWT | PLMA |
|---|---|---|---|---|
| **Name** | Gap / Time | Gap / Time | Gap / Time | Gap / Time |
| tho30 | (0.00, **0.00**, 0.00) / 0.05 | (0.00, 0.32, 0.57) / 0.54 | (22.27, 25.44, 29.29) / 13.67 | (0.00, 0.03, 0.29) / 1.58 |
| tho40 | (0.00, **0.02**, 0.05) / 2.55 | (0.14, 0.41, 0.89) / 0.84 | (22.31, 28.52, 30.38) / 13.93 | (0.00, 0.04, 0.20) / 4.04 |
| wil100 | (0.01, **0.04**, 0.09) / 45.72 | (0.07, 0.16, 0.21) / 16.54 | (7.80, 8.17, 8.46) / 19.51 | (0.01, 0.04, 0.08) / 7.42 |
| wil50 | (0.00, **0.01**, 0.03) / 3.33 | (0.02, 0.07, 0.16) / 2.79 | (9.11, 9.86, 10.39) / 14.25 | (0.00, 0.01, 0.05) / 3.54 |

Table 9: Detailed comparison on Taixxeyy instances (10 repetitions). Each cell shows (min/mean/max) Gap (%) and average Time (s).

| Problem | Ro-TS | IPFP | PLMA |
|---|---|---|---|
| **Name** | (min/mean/max) Gap / Time | (min/mean/max) Gap / Time | (min/mean/max) Gap / Time |
| tai27e01 | (0.00, 7.58, 19.16) / 0.62 | (8.44, 15.86, 22.67) / 0.36 | (-0.00, **-0.00**, -0.00) / 0.14 |
| tai27e02 | (0.00, 0.25, 0.84) / 0.56 | (6.25, 15.96, 26.53) / 0.33 | (-0.00, **-0.00**, -0.00) / 0.08 |
| tai27e03 | (0.00, 2.68, 14.18) / 0.66 | (17.62, 24.44, 31.92) / 0.38 | (0.00, **0.00**, 0.00) / 0.07 |
| tai27e04 | (0.00, 89.99, 405.32) / 0.71 | (12.40, 20.96, 33.88) / 0.41 | (0.00, **0.00**, 0.00) / 0.08 |
| tai27e05 | (2.15, 50.76, 433.70) / 0.82 | (3.77, 17.15, 34.94) / 0.37 | (-0.00, **-0.00**, -0.00) / 0.08 |
| tai27e06 | (0.00, 40.48, 371.00) / 0.54 | (5.69, 23.73, 35.18) / 0.39 | (0.00, **0.00**, 0.00) / 0.07 |
| tai27e07 | (0.00, 6.60, 14.88) / 0.60 | (10.33, 22.32, 44.87) / 0.38 | (-0.00, **-0.00**, -0.00) / 0.07 |
| tai27e08 | (0.00, 147.98, 492.43) / 0.41 | (10.12, 31.15, 59.34) / 0.42 | (0.00, **0.00**, 0.00) / 0.07 |
| tai27e09 | (0.00, 69.25, 337.28) / 0.49 | (3.31, 19.33, 33.49) / 0.39 | (0.00, **0.00**, 0.00) / 0.08 |
| tai27e10 | (0.00, 44.54, 438.41) / 0.36 | (12.22, 23.15, 37.07) / 0.39 | (-0.00, **-0.00**, -0.00) / 0.08 |
| tai27e11 | (0.00, 41.68, 382.66) / 0.63 | (1.38, 18.87, 27.80) / 0.40 | (0.00, **0.00**, 0.00) / 0.06 |
| tai27e12 | (0.00, 2.69, 11.40) / 0.67 | (2.02, 16.74, 31.47) / 0.35 | (-0.00, **-0.00**, -0.00) / 0.08 |
| tai27e13 | (0.00, 158.72, 390.63) / 0.59 | (0.40, 13.43, 27.85) / 0.39 | (0.00, **0.00**, 0.00) / 0.07 |
| tai27e14 | (0.00, 3.42, 10.48) / 0.45 | (4.65, 15.76, 32.40) / 0.39 | (0.00, **0.00**, 0.00) / 0.07 |
| tai27e15 | (0.00, 3.54, 16.36) / 0.67 | (15.98, 21.71, 39.19) / 0.37 | (0.00, **0.00**, 0.00) / 0.09 |
| tai27e16 | (0.00, 34.65, 346.48) / 0.43 | (0.00, 19.74, 31.31) / 0.38 | (-0.00, **-0.00**, -0.00) / 0.05 |
| tai27e17 | (0.00, 29.34, 293.39) / 0.26 | (0.26, 16.54, 30.05) / 0.38 | (0.00, **0.00**, 0.00) / 0.08 |
| tai27e18 | (0.00, 85.84, 419.14) / 0.61 | (0.00, 16.15, 25.89) / 0.37 | (-0.00, **-0.00**, -0.00) / 0.08 |
| tai27e19 | (0.00, 6.21, 15.27) / 0.76 | (4.61, 14.08, 31.98) / 0.37 | (0.00, **0.00**, 0.00) / 0.08 |
| tai27e20 | (0.00, 3.75, 8.49) / 0.51 | (3.34, 22.24, 44.28) / 0.39 | (0.00, **0.00**, 0.00) / 0.07 |
| tai45e01 | (7.30, 58.87, 382.75) / 3.89 | (11.48, 24.05, 47.07) / 0.97 | (0.00, **0.00**, 0.00) / 0.25 |
| tai45e02 | (1.33, 60.63, 401.19) / 3.89 | (16.92, 24.29, 32.68) / 1.06 | (-0.00, **-0.00**, -0.00) / 0.24 |
| tai45e03 | (0.11, 202.98, 486.07) / 3.89 | (1.02, 19.10, 34.07) / 1.06 | (0.00, **0.00**, 0.00) / 0.23 |
| tai45e04 | (0.84, 136.56, 446.88) / 3.90 | (16.36, 30.26, 49.66) / 1.04 | (-0.00, **0.70**, 6.99) / 0.74 |
| tai45e05 | (1.62, 181.32, 445.89) / 3.89 | (11.25, 28.66, 44.38) / 1.06 | (-0.00, **-0.00**, -0.00) / 0.27 |
| tai45e06 | (0.51, 56.11, 467.88) / 3.90 | (6.05, 16.23, 29.89) / 1.03 | (0.00, **0.00**, 0.00) / 0.17 |
| tai45e07 | (0.43, 72.06, 347.09) / 3.90 | (2.07, 21.99, 37.52) / 1.01 | (0.00, **0.00**, 0.00) / 0.86 |
| tai45e08 | (1.43, 156.65, 371.93) / 3.90 | (2.93, 22.55, 37.90) / 1.01 | (0.00, **0.00**, 0.00) / 0.19 |
| tai45e09 | (1.87, 102.48, 486.58) / 3.89 | (7.04, 22.74, 36.19) / 0.99 | (0.00, **0.00**, 0.00) / 0.25 |
| tai45e10 | (0.02, 154.62, 380.81) / 3.89 | (1.83, 17.21, 32.46) / 1.00 | (-0.00, **-0.00**, -0.00) / 0.23 |
| tai45e11 | (0.00, 51.49, 425.75) / 3.74 | (12.50, 20.74, 37.36) / 1.12 | (0.00, **0.00**, 0.00) / 0.18 |
| tai45e12 | (0.00, 51.22, 414.57) / 3.68 | (9.19, 19.29, 38.46) / 0.99 | (-0.00, **-0.00**, -0.00) / 0.32 |
| tai45e13 | (0.00, 85.56, 381.99) / 3.60 | (10.75, 24.52, 43.82) / 1.03 | (0.00, **0.00**, 0.00) / 0.19 |
| tai45e14 | (0.00, 19.08, 87.45) / 3.80 | (16.25, 25.77, 36.24) / 1.07 | (0.00, **0.00**, 0.00) / 0.19 |
| tai45e15 | (0.00, 126.79, 401.33) / 3.86 | (1.43, 21.44, 31.57) / 1.09 | (0.00, **0.00**, 0.00) / 0.18 |
| tai45e16 | (0.37, 42.02, 374.85) / 3.89 | (0.95, 19.01, 42.52) / 1.00 | (0.00, **0.00**, 0.00) / 0.27 |
| tai45e17 | (2.18, 98.03, 300.45) / 3.89 | (0.23, 12.26, 25.89) / 1.02 | (0.00, **0.00**, 0.00) / 0.25 |
| tai45e18 | (0.00, 87.40, 413.12) / 3.58 | (7.96, 21.26, 33.48) / 1.02 | (0.00, **0.00**, 0.00) / 0.21 |
| tai45e19 | (0.00, 172.91, 429.12) / 3.64 | (18.38, 24.73, 33.42) / 1.02 | (0.00, **0.00**, 0.00) / 0.14 |
| tai45e20 | (2.09, 121.06, 566.27) / 3.89 | (10.51, 24.10, 35.02) / 1.00 | (-0.00, **-0.00**, -0.00) / 0.15 |
| tai75e01 | (9.73, 95.31, 283.74) / 18.47 | (20.49, 29.09, 36.29) / 2.60 | (-0.00, **-0.00**, -0.00) / 1.69 |
| tai75e02 | (11.80, 99.41, 294.86) / 18.50 | (24.92, 31.31, 42.26) / 2.53 | (-0.00, **0.42**, 2.08) / 2.43 |
| tai75e03 | (0.81, 67.62, 272.54) / 18.48 | (20.15, 28.83, 39.44) / 2.59 | (0.00, **0.00**, 0.00) / 0.91 |
| tai75e04 | (2.66, 145.42, 285.63) / 18.49 | (19.00, 27.71, 37.71) / 2.58 | (-0.00, **-0.00**, -0.00) / 1.49 |

| Problem | Ro-TS | IPFP | PLMA |
|---|---|---|---|
| **Name** | (min/mean/max) Gap / Time | (min/mean/max) Gap / Time | (min/mean/max) Gap / Time |
| tai75e05 | (3.15, 89.91, 278.31) / 18.49 | (17.11, 24.36, 35.36) / 2.40 | (-0.00, **0.08**, 0.84) / 1.33 |
| tai75e06 | (3.32, 48.85, 311.87) / 18.48 | (23.17, 40.10, 47.44) / 2.73 | (-0.00, **-0.00**, -0.00) / 1.35 |
| tai75e07 | (8.78, 196.51, 324.15) / 18.48 | (19.78, 26.45, 36.98) / 2.38 | (0.00, **0.28**, 0.93) / 3.18 |
| tai75e08 | (10.96, 76.87, 293.53) / 18.49 | (14.51, 26.28, 36.46) / 2.27 | (-0.00, **-0.00**, -0.00) / 1.26 |
| tai75e09 | (1.58, 113.32, 275.30) / 18.48 | (19.83, 26.90, 33.08) / 2.56 | (0.00, **0.00**, 0.00) / 1.59 |
| tai75e10 | (5.90, 149.17, 288.37) / 18.47 | (14.01, 22.03, 34.58) / 2.47 | (-0.00, **-0.00**, -0.00) / 0.94 |
| tai75e11 | (3.27, 124.39, 294.39) / 18.47 | (13.73, 26.45, 37.61) / 2.57 | (-0.00, **-0.00**, -0.00) / 1.04 |
| tai75e12 | (12.48, 154.06, 295.78) / 18.47 | (25.02, 30.11, 34.25) / 2.41 | (-0.00, **0.41**, 2.04) / 2.68 |
| tai75e13 | (5.99, 90.61, 282.66) / 18.48 | (11.39, 25.02, 38.85) / 2.56 | (0.00, **0.63**, 3.15) / 1.76 |
| tai75e14 | (4.25, 19.70, 49.03) / 18.48 | (20.41, 29.04, 36.85) / 2.57 | (-0.00, **-0.00**, -0.00) / 1.33 |
| tai75e15 | (3.47, 145.10, 286.22) / 18.50 | (17.24, 27.54, 33.59) / 2.59 | (0.00, **0.00**, 0.00) / 0.88 |
| tai75e16 | (19.85, 207.84, 292.96) / 18.50 | (16.56, 22.94, 30.36) / 2.44 | (-0.00, **-0.00**, -0.00) / 1.60 |
| tai75e17 | (2.29, 140.05, 272.35) / 18.47 | (7.93, 27.85, 42.62) / 2.63 | (0.00, **0.00**, 0.00) / 1.06 |
| tai75e18 | (8.33, 70.83, 296.13) / 18.48 | (17.60, 27.43, 35.41) / 2.58 | (0.00, **0.00**, 0.00) / 0.92 |
| tai75e19 | (4.38, 148.63, 332.59) / 18.51 | (16.17, 29.39, 36.27) / 2.34 | (0.00, **0.00**, 0.00) / 1.14 |
| tai75e20 | (1.11, 41.99, 289.84) / 18.51 | (12.16, 23.94, 30.26) / 2.50 | (-0.00, **-0.00**, -0.00) / 0.93 |
| tai125e01 | (11.43, 45.68, 298.57) / 72.52 | (17.79, 27.42, 33.23) / 9.35 | (0.10, **2.36**, 3.74) / 9.92 |
| tai125e02 | (6.92, 106.10, 318.56) / 72.58 | (21.58, 25.23, 30.64) / 9.32 | (1.13, **3.38**, 5.58) / 8.53 |
| tai125e03 | (8.68, 76.74, 308.06) / 72.48 | (21.30, 29.87, 36.27) / 9.19 | (3.26, **6.02**, 9.83) / 8.58 |
| tai125e04 | (1.94, 11.68, 20.43) / 72.53 | (23.30, 29.17, 38.79) / 9.47 | (-0.48, **2.48**, 4.84) / 8.16 |
| tai125e05 | (-0.04, 60.94, 272.41) / 72.41 | (19.70, 24.75, 31.86) / 8.00 | (-0.26, **2.08**, 5.53) / 7.64 |
| tai125e06 | (11.51, 98.77, 292.30) / 72.50 | (19.27, 24.05, 28.30) / 7.90 | (-0.73, **5.30**, 8.11) / 9.75 |
| tai125e07 | (10.22, 46.77, 284.53) / 72.45 | (21.61, 30.61, 34.91) / 8.06 | (-1.81, **4.64**, 9.37) / 9.02 |
| tai125e08 | (9.30, 112.84, 265.66) / 72.50 | (16.88, 24.99, 31.88) / 8.28 | (-1.05, **0.24**, 1.60) / 6.78 |
| tai125e09 | (10.25, 72.13, 300.55) / 72.52 | (23.70, 27.89, 33.86) / 9.02 | (-0.08, **2.34**, 4.30) / 7.73 |
| tai125e10 | (11.51, 106.42, 302.24) / 72.59 | (20.32, 27.32, 33.29) / 8.48 | (0.27, **2.36**, 5.40) / 8.24 |
| tai125e11 | (4.66, 118.39, 284.63) / 72.60 | (19.76, 27.42, 32.06) / 8.38 | (0.14, **3.39**, 8.96) / 8.60 |
| tai125e12 | (16.09, 111.32, 317.21) / 72.53 | (23.71, 29.60, 38.10) / 8.02 | (1.04, **3.42**, 7.33) / 8.43 |
| tai125e13 | (6.73, 117.73, 273.36) / 72.61 | (24.20, 28.02, 34.10) / 7.82 | (-0.11, **1.89**, 5.57) / 7.96 |
| tai125e14 | (7.23, 100.60, 299.06) / 72.55 | (22.97, 32.20, 36.79) / 9.16 | (-0.09, **3.07**, 6.08) / 8.31 |
| tai125e15 | (5.90, 125.11, 300.54) / 72.50 | (19.74, 25.21, 29.43) / 8.47 | (-2.00, **0.26**, 3.46) / 6.57 |
| tai125e16 | (3.94, 114.62, 269.72) / 72.52 | (12.41, 18.97, 23.78) / 8.21 | (0.12, **2.64**, 6.71) / 10.21 |
| tai125e17 | (3.49, 17.88, 34.90) / 72.49 | (23.08, 26.79, 31.88) / 8.30 | (1.79, **3.13**, 5.75) / 8.38 |
| tai125e18 | (5.01, 84.94, 264.67) / 72.54 | (16.86, 22.36, 29.61) / 8.17 | (-0.83, **0.17**, 1.65) / 5.74 |
| tai125e19 | (10.28, 74.85, 301.27) / 72.51 | (26.38, 30.59, 34.97) / 8.73 | (0.03, **4.20**, 7.52) / 8.41 |
| tai125e20 | (8.04, 47.04, 302.06) / 72.50 | (18.23, 26.13, 29.70) / 8.10 | (-1.98, **-0.04**, 1.08) / 6.58 |
| tai175e01 | (8.04, 136.55, 265.64) / 158.69 | (19.22, 24.06, 32.44) / 17.62 | (5.31, **9.11**, 13.11) / 14.26 |
| tai175e02 | (9.88, 47.83, 277.88) / 158.74 | (18.68, 25.77, 30.26) / 16.58 | (9.87, **12.11**, 14.19) / 14.34 |
| tai175e03 | (15.32, 164.24, 305.14) / 158.76 | (23.00, 27.93, 32.75) / 17.55 | (5.51, **8.86**, 13.28) / 14.35 |
| tai175e04 | (7.00, 60.56, 248.45) / 158.69 | (9.38, 17.93, 22.35) / 17.09 | (4.37, **7.11**, 8.72) / 14.31 |
| tai175e05 | (13.93, 52.22, 297.51) / 158.71 | (24.95, 27.94, 31.82) / 16.35 | (6.92, **9.87**, 11.75) / 15.11 |
| tai175e06 | (11.32, 42.79, 272.55) / 158.70 | (21.06, 25.38, 30.60) / 16.68 | (5.90, **8.60**, 10.03) / 15.39 |
| tai175e07 | (11.09, 71.72, 279.85) / 158.75 | (22.78, 25.49, 28.59) / 16.58 | (8.96, **10.38**, 12.88) / 14.20 |
| tai175e08 | (6.15, 40.32, 262.51) / 158.65 | (16.25, 22.98, 26.68) / 16.51 | (8.16, **13.89**, 17.99) / 15.32 |
| tai175e09 | (-0.03, 57.54, 248.44) / 147.71 | (13.23, 18.37, 22.17) / 16.75 | (-1.12, **9.58**, 14.80) / 13.01 |
| tai175e10 | (7.47, 49.93, 270.09) / 158.71 | (17.61, 24.59, 28.04) / 17.05 | (9.25, **10.41**, 11.75) / 14.48 |
| tai175e11 | (10.39, 93.45, 270.99) / 158.84 | (17.87, 22.62, 29.26) / 16.59 | (9.47, **12.45**, 14.99) / 15.89 |
| tai175e12 | (11.51, 42.96, 263.97) / 158.79 | (11.86, 20.83, 27.91) / 16.80 | (1.36, **5.17**, 8.94) / 14.47 |
| tai175e13 | (10.38, 17.83, 29.28) / 158.69 | (18.63, 22.93, 27.17) / 16.71 | (7.07, **10.40**, 13.18) / 14.69 |
| tai175e14 | (2.43, 83.00, 255.31) / 158.63 | (9.50, 20.07, 27.54) / 16.71 | (3.32, **5.84**, 10.07) / 14.31 |
| tai175e15 | (13.36, 43.07, 274.57) / 158.77 | (18.11, 22.42, 30.63) / 17.21 | (4.39, **8.62**, 12.44) / 14.94 |
| tai175e16 | (7.68, 106.06, 311.13) / 158.80 | (19.09, 25.37, 30.55) / 16.85 | (5.56, **8.39**, 11.43) / 14.29 |
| tai175e17 | (8.60, 65.70, 266.28) / 158.71 | (18.13, 21.64, 23.99) / 16.53 | (3.56, **5.25**, 7.69) / 15.19 |
| tai175e18 | (12.11, 42.13, 266.07) / 158.66 | (19.35, 22.05, 24.82) / 17.27 | (4.40, **6.28**, 7.89) / 15.54 |
| tai175e19 | (6.29, 95.24, 271.39) / 158.82 | (17.88, 23.23, 27.26) / 16.66 | (4.50, **9.01**, 11.39) / 14.34 |
| tai175e20 | (9.31, 44.14, 282.49) / 158.73 | (17.47, 24.71, 27.46) / 17.13 | (5.34, **10.90**, 14.24) / 14.20 |

# E    THE USE OF LARGE LANGUAGE MODELS

Large Language Models (LLM) is used only for polishing the writing.

