# OpenReview forum: "Learning the Quadratic Assignment Problem with Warm-Started MCMC Finetuning and Cross-Graph Attention"
_ICLR.cc/2026/Conference — Submitted to ICLR 2026_

### Official Review · Reviewer_M7LG · 2025-10-26

**Soundness:** 3
**Presentation:** 3
**Contribution:** 3
**Rating:** 6
**Confidence:** 3

**Summary:**

This work proposed a probabilistic framework for learning QAP problems, which is proved in the paper that is equivalent to the original deterministic model. For the modeling of the distributions, the authors proposed an instance-based model that takes an instance as input and outputs the parameterized distribution of the assignment heatmap matrix. The model architecture is a GNN based on two graphs of the problem, with attention mechanism across the graphs. This framework thus allows pretraining and deployment time finetuning.

**Strengths:**

- Overall, the paper is well written, and the overall framework is well motivated.
- The probabilistic approach for NP-hard QAP problems is novel and prospective.
- The MCMC sampling is well designed and is a good approach for intractable distributions.
- The empirical results are promising in terms of both time and gap, as well as generalization on larger problems.

**Weaknesses:**

- I suggest adding an overall illustration of the framework, so that the concept is more straightforward on first seeing it.
- Some concepts and details are not well explained to me, see my questions below.

**Questions:**

- For learning based methods on CO problems, is it common to do deployment time finetuning on instances?
- In equation 6, I still don't understand why $||\phi||_F^2$ term can be eliminated. Isn't it the output of the neural networks, and should be considered as well?
- As far as I understand, the node embeddings $h_{ini}$ are the same for all nodes? And the distinction across instances are in $D$ and $F$? But what exactly are $D$ and $F$? I think it is important but i didn't find where $D$ and $F$ are defined. Besides, how do I interpret $\bar{D}$ and $\bar{F}$?
- In equation 12, why do we need $C$ term and how is it tuned?
- In line 253: "This structure actually yields a low-rank approximation...." why is it low rank?
- In e.g. Table 1, what is the reference solution based on which the gap is calculated?
- In Figure 1, we can see the ablation that without the attention it is worse, but what is the time overhead with the attention? If it is a lot of extra time then maybe removing the attention may also be acceptable.
- In training as well finetuning, what is intuitively the $\hat{G}$? And how is the gradient $\nabla_{\theta} \Phi_{\theta}$ obtained?
- Finally, could you explain more on the pushforward transformation design? I can see it is to smooth the loss, but I don't get it how it is done. And it is also confusing what is conveyed in appendix B.1.

---

> ### Author Response · Authors · 2025-11-25
> **Response to Reviewer M7LG (1/3)**
>
> We sincerely thank the reviewer for the positive assessment. We have carefully addressed your specific questions below to further clarify our work.
>
> > **W1. I suggest adding an overall illustration of the framework, so that the concept is more straightforward on first seeing it.**
>
> We thank the reviewer for this helpful suggestion. We agree that an overall illustration would provide a more intuitive understanding of our framework, given its hybrid nature of combining neural parameterization with MCMC-based finetuning. To clarify the workflow, our method operates through three integrated stages:
>
> * **Cross-Graph Encoding**: The model takes the Flow ($F$) and Distance ($D$) matrices as inputs. Instead of constructing a large association graph, we employ separate GNNs coupled with a Cross-Graph Attention mechanism to directly fuse information between the two graphs, producing a heatmap that captures structural dependencies.
> * **EBM Parameterization:** This heatmap parametrizes an Energy-Based Model (EBM) with an additive structure. This design allows for $O(1)$ cost evaluation of 2-swap updates, enabling highly efficient exploration of the permutation space.
> * **Warm-Started MCMC Finetuning:** The inference process is an iterative loop. We employ Batched Warm-Started MCMC, where high-quality solutions from the previous iteration are reused to initialize short parallel Markov chains. These chains are refined via local improvement and evaluated to estimate policy gradients, which finetune the encoder on-the-fly to guide the search toward promising regions.
>
> > **Q1. For learning based methods on CO problems, is it common to do deployment time finetuning on instances?**
>
> Yes, as discussed in our Related Work section, it is indeed a common practice for learning-based CO methods to perform finetuning directly on the test instances during deployment. Active search [1] is a pioneering paradigm in this area, where model parameters are optimized specifically for the target test instances. Variants include EAS [2], which updates only a subset of parameters or embeddings for efficiency, and DIMES [3], which employs meta-learning to provide favorable initialization for finetuning. Furthermore, many recent works utilize EAS either as a strong baseline or as a component to enhance performance [4,5].
>
> These active search methods are generally built upon auto-regressive models, which factorize the overall probability into a product of conditional probabilities and construct solutions sequentially in an element-wise manner. However, this process requires restarting from scratch at each refinement iteration, hindering effective reuse of previously identified high-quality solutions. In contrast, our work introduces a EBM with MCMC sampling, enabling warm-starts from prior solutions and overcoming the restart limitation of auto-regressive active search.
>
> > **Q2. In equation 6, I still don't understand why $\frac{1}{2}||\phi||^2_F$ term can be eliminated. Isn't it the output of the neural networks, and should be considered as well?**
>
> We use $\propto$ (proportionality) in Eq. 6 as we are defining a probabilistic model over permutations. Terms that are constant with respect to the permutation $\pi$ (such as $\frac{1}{2}\|\phi\|^2_F$ and $\frac{1}{2}\|X_{\pi}\|^2_F$) are absorbed into the normalization constant. This constant cancels out in the MCMC acceptance ratio, so it can be omitted.
>
> > **Q3. As far as I understand, the node embeddings $h\_{ini}$ are the same for all nodes? And the distinction across instances are in $D$ and $F$ ? But what exactly are $D$ and $F$? I think it is important but i didn't find where $D$ and $F$ are defined. Besides, how do I interpret $\bar{D}$ and $\bar{F}$ ?**
>
> You are correct that $h_{ini}$ is a learnable initial parameter shared by all nodes, serving as a starting point for the embeddings. The instance-specific structural information is entirely injected via the matrices $F$ and $D$. As defined in Eq. 1 & 2 (Section 2.1), $F$ represents the Flow matrix (flows between facilities) and $D$ represents the Distance matrix (distances between locations). These two matrices fundamentally define a Koopmans-Beckmann QAP instance. $\bar{D}$ and $\bar{F}$ represent the scalar averages of the elements in matrices $F$ and $D$, respectively. In Eq. 9, we subtract these means (e.g., $D - \bar{D}$) as a centering operation and this performs well in practice.
>
> > **Q4. In equation 12, why do we need term and how is it tuned?**
>
> $C$ is a clipping hyperparameter that scales the output of $\tanh$ function from $(-1, 1)$ to $(-C, C)$. This helps control the entropy or "temperature" of the resulting permutation distribution. This practice is similar with prior works like [6] and [7]. In practice, we empirically select $C$ in different scenarios.

---

> ### Author Response · Authors · 2025-11-25
> **Response to Reviewer M7LG (2/3)**
>
> > **Q5. In line 253: "This structure actually yields a low-rank approximation...." why is it low rank?**
>
> We refer to the "Low-Rank Permutation Matrix Representation" in [8]. Mathematically, the final output in Eq. 12 is not low-rank. But before the non-linear activation, the term $\tilde{H}_F (\tilde{H}_D)^T$ is the multiplication of two $n\times d$ matrices, where $d$ is the embedding dimension and $n$ is the problem size. When $d$ is fixed and $n$ grows, it's effectively a "low-rank approximation".
>
> > **Q6. In e.g. Table 1, what is the reference solution based on which the gap is calculated?**
>
> We apologize for the lack of clarity. As noted briefly  in the _Metrics_ section, the reference solutions are the best known solutions for benchmarks (QAPLIB, Taixxeyy) and the Ro-TS solutions for synthetic datasets.
>
> > **Q7. In Figure 1, we can see the ablation that without the attention it is worse, but what is the time overhead with the attention? If it is a lot of extra time then maybe removing the attention may also be acceptable.**
>
> Thank you for your question. You are correct to consider the trade-off between performance and computational cost. In our full model, the single cross-attention block at the final stage accounts for approximately 20-25% of the total inference time, which we consider a reasonable cost. More importantly, our ablation study in Figure 1 was designed to provide a direct and fair comparison. When we created the "without attention" version, we didn't just remove the module. Instead, we added more GNN layers to it. This step ensured that both models, our full model and the one without attention, have a very similar number of parameters and take almost the same inference times. Therefore, the drop in performance shown in the figure is not because the model became smaller or simpler. It directly shows that, for the same computational cost, our architecture with the attention module works much better than an alternative that just uses a deeper GNN. This result strongly justifies including the attention mechanism in our design.
>
> > **Q8. In training as well finetuning, what is intuitively the $\hat{G}$ ? And how is the gradient $\nabla_\theta \Phi_{\theta}$ obtained?**
>
> Intuitively, $\hat{G}$ acts similarly to the policy gradient in Reinforcement Learning. The term $(g(\pi_i) - b)\nabla_\theta \Phi_\theta(\pi_i)$ encourages the model to increase the "energy" (or unnormalized log-probability) $\Phi_\theta$ of sampled permutations $\pi_i$ that have a cost $g(\pi_i)$ lower than the baseline $b$.
>
> Technically, $\Phi_\theta(\pi)$ is the output score of our neural network for a given permutation. Since the network is a standard differentiable computation graph (PyTorch model), $\nabla_\theta \Phi_\theta(\pi)$ is obtained simply by performing standard backpropagation on the network output $\Phi_\theta$. We do not need to differentiate through the sampling process itself, only through the score function parameterization.
>
> > **Q9. Finally, could you explain more on the pushforward transformation design? I can see it is to smooth the loss, but I don't get it how it is done. And it is also confusing what is conveyed in appendix B.1.**
>
> We clarify the implementation and the "smoothing" effect as follows:
>
> * **Implementation of the Pushforward Transformation.** The implementation directly follows Equation (5) in the paper. Instead of explicitly constructing a probability distribution, we integrate the transformation into the training loop: (i) Sample: Draw a permutation $\pi$ from the network $p_\theta$. (ii) Refine: Apply the local search operator $\mathcal{T}$ (detailed in Appendix B.1) to obtain $\pi' = \mathcal{T}(\pi)$. (iii) Update: Compute the gradient using the cost of the refined solution $f(\pi')$.
> * **How it smooths the loss.** The QAP objective landscape is discrete and rugged, where high-quality solutions are sparse isolated points. The operator $\mathcal{T}$ maps a broad set of neighboring permutations to the same local optimum. By optimizing the cost after this mapping, we effectively group these neighbors together. This expands the target from a single optimal point to a larger "promising region." The network only needs to generate a sample falling anywhere within this region to achieve a low loss, resulting in a much flatter and easier-to-optimize landscape.
> * **Connection to Appendix B.1.** Appendix B.1 (Algorithm 2) is the concrete definition of $\mathcal{T}$. It specifies the exact greedy 2-swap search procedure used to refine the sample $\pi$ into $\pi'$ during the "Refine" step mentioned above.

---

> ### Author Response · Authors · 2025-11-25
> **Response to Reviewer M7LG (3/3)**
>
> [1] Irwan Bello, Hieu Pham, Quoc V Le, Mohammad Norouzi, and Samy Bengio. "Neural combinatorial optimization with reinforcement learning." ICLR, 2017.
> [2] Andr´e Hottung, Yeong-Dae Kwon, and Kevin Tierney. "Efficient active search for combinatorial optimization problems." ICLR, 2022.
> [3] Ruizhong Qiu, Zhiqing Sun, and Yiming Yang. "DIMES: A differentiable meta solver for combinatorial optimization problems." NeurIPS, 2022.
> [4] Choo, Jinho, et al. "Simulation-guided beam search for neural combinatorial optimization." NeurIPS, 2022.
> [5] Sun, Zhiqing, and Yiming Yang. "Difusco: Graph-based diffusion solvers for combinatorial optimization." NeurIPS, 2023.
>
> [6] Irwan Bello, Hieu Pham, Quoc V Le, Mohammad Norouzi, and Samy Bengio. Neural combinatorial optimization with reinforcement learning. _arXiv preprint arXiv:1611.09940_, 2016.
>
> [7] Wouter Kool, Herke Van Hoof, and Max Welling. Attention, learn to solve routing problems! In _International Conference on Learning Representations_, 2018.
>
> [8] Hannah Dr¨oge, Zorah L¨ahner, Yuval Bahat, Onofre Martorell Nadal, Felix Heide, and Michael Moeller. Kissing to find a match: Efficient low-rank permutation representation. In Advances in Neural Information Processing Systems 36: Annual Conference on Neural Information Processing Systems 2023, NeurIPS 2023, New Orleans, LA, USA, December 10 - 16, 2023, 2023.

---

> > ### Comment · Reviewer_M7LG · 2025-11-25
> >
> > Thank you for your effort and your detailed response!
> > The clarification regarding my concerns and questions are all solved. However, by reading other reviewer's comment, I have to acknowledge there are still some weaknesses of the paper, e.g., lack of comparisons to prior work or strong heuristics, even though there might be some reasons like no open source code preventing you doing this.
> > I remain positive about the work, especially the novelty of the overall framework and the MCMC sampling design. After consideration, I would like to keep my score, I believe it is fair judgement.

---

### Official Review · Reviewer_6tz8 · 2025-10-29

**Soundness:** 2
**Presentation:** 3
**Contribution:** 2
**Rating:** 4
**Confidence:** 4

**Summary:**

The authors present a QAP solver based on the following. Pretraining minimizes the expected post-improvement cost via a pushforward map using a prior model parameterized as an energy-based model (EBM).  Fine-tuning starts from the pretrained model, runs the pushforward map, updates the EBM parameters, and returns the best solution found. The energy of a permutation is computed from a compatibility heatmap parameterized by a cross-graph attention network.

**Strengths:**

Well-structured paper with extensive benchmarks and clear awareness of recent developments in neuro optimization. Attention and graph-matching networks are not new per se; the novelty lies in their combination with the EBM and sampling framework. Main achievement seems to be competitive performance against a solid (although not champion) heuristic algorithm and better performance than previously proposed in its subfield.

**Weaknesses:**

Novelty is limited beyond the combination of existing methods. Benchmarking is narrow with respect to classical heuristics (Ro-TS alone may not suffice). Ablation studies are not entirely clear. Improvements appear incremental, and it is hard to assess their significance. Limited analysis of the contribution of pretraining.

**Questions:**

1) On what hardware was Ro-TS run? Were all runs performed on GPU? Was Ro-TS executed on CPU? Since the reported gap is similar to Ro-TS, runtime comparisons are only meaningful relative to it.

2) Are there stronger baselines (e.g., ITS, ILS)?
See for example:
- Misevičius, A. Letter: New best known solution for the most difficult QAP instance “tai100a.” Memetic Computing 11, 331–332 (2019).
- Misevičius, A. An implementation of the iterated tabu search algorithm for the quadratic assignment problem. OR Spectrum 34(3): 665–690 (2012).
- Hussin, M. S., & Stützle, T. Hierarchical iterated local search for the quadratic assignment problem. Hybrid Metaheuristics Workshop, Springer, 2009.

3) What are the “diverse instances” used for pretraining? Specify distributions. Any overlap with test families?

4) Why is the ablation study conducted only on the uniform dataset? No error bars are provided so the relative differences are not meaningful.

5) What about memory overhead of the proposed method?

6) Emphasizing time to reach the same or better gap as SOTA would be clearer? Mixed tables of gap and runtime are harder to interpret. Time-to-target (TTT) metrics are standard and more informative.

7) The key novelty is learning-based improvement. How much pretraining helps compared to classical heuristics by examining generalization for in-distribution and out-of-distribution instances?

---

> ### Author Response · Authors · 2025-11-25
> **Response to Reviwer  6tz8 (1/4)**
>
> We appreciate the reviewer’s detailed comments and rigorous attention to our evaluation protocols. We have prepared the following responses to resolve your questions.
>
> > **Q1. On what hardware was Ro-TS run? Were all runs performed on GPU? Was Ro-TS executed on CPU? Since the reported gap is similar to Ro-TS, runtime comparisons are only meaningful relative to it.**
>
> We confirm that Ro-TS was executed on CPU (Intel Xeon Gold 6326), while PLMA was executed on GPU (NVIDIA Tesla A800). However, we argue that the runtime gap is not merely a result of hardware capability, but stems from the fundamental algorithmic difference: Ro-TS is an inherently serial "deep" search, whereas PLMA is a naturally parallel "broad" search. PLMA’s design is explicitly tailored to unlock the potential of modern parallel hardware, which is a key contribution of our neural framework.
>
> Ro-TS operates on a single solution trajectory. It requires a long chain of sequential updates (e.g., $5000 \times n$ iterations), where iteration $t+1$ strictly depends on the result of iteration $t$. This temporal dependency makes it extremely difficult to parallelize the outer loop effectively on GPUs.
>
> In contrast, PLMA is designed to be parallel within the solution space. Instead of one long chain, we distribute the search across $K \times M = 400$ independent short chains. This design allows us to map the computational workload perfectly onto GPU threads.
>
> To demonstrate that our efficiency comes from the algorithm structure, we compare the number of 2-swap evaluations performed by a single chain in PLMA versus the single chain in Ro-TS:
>
> * Ro-TS (Total/Single chain): $\approx 5000n \times \frac{n(n-1)}{2}$
> * PLMA (Total): $\approx 200 \times 400 \times n \times 2n$
> * Ratio per chain: Since PLMA distributes the total effort across 400 independent chains, the sequential computational burden for a single chain is:
>
> $$
> \text{Ratio}\_{\text{per-chain}} = \frac{N_{\\text{PLMA-total}} / 400}{N_{\\text{RoTS}}} \approx \frac{0.16}{n-1}.
> $$
>
> For $n=100$, a single PLMA chain performs only ~0.16% of the sequential evaluations required by Ro-TS. This drastic reduction in per-chain depth, combined with the parallel execution of 400 chains (sample parallelism) and 256 instances (batch parallelism), is the root cause of PLMA's superior efficiency.

---

> ### Author Response · Authors · 2025-11-25
> **Response to Reviwer  6tz8 (2/4)**
>
> > **Q2. Are there stronger baselines (e.g., ITS, ILS)?**
>
> We appreciate the reviewer for suggesting advanced heuristics like ITS and ILS [1, 2, 3] to broaden our benchmarking. While we acknowledge their potential, we maintained Ro-TS as our primary strong baseline for three reasons:
>
> * **Reproducibility**: We prioritized baselines with official, publicly available code to ensure reproducibility. Ro-TS has a robust standard C implementation, whereas the specific ITS/ILS variants lack official code. As shown in our re-implementation attempts (Table R4-R6), achieving the exact reported performance without original codebase is challenging due to the sensitivity of heuristics to undocumented implementation nuances. To prevent misleading results from a potentially sub-optimal re-implementation, we adopted the official Ro-TS to ensure a fair evaluation
> * **Competitiveness of Our Ro-TS Baseline**: We used a "High-Effort" Ro-TS configuration (5000 $\\times$ $n$ iterations on Intel Xeon Gold 6326). Under this setting, the gap is minimal. For instance, on `sko100a`, our Ro-TS achieves a mean gap of 0.08%, which is comparable to the 0.09% reported in the ITS reference. The narrow margin between our high-effort Ro-TS and the reported ITS results confirms that the official Ro-TS, when run with sufficient resources, serves as a formidable baseline for evaluating our PLMA solver.
> * **Raised Benchmarking Standards**: Prior learning-based works (e.g., NGM, SAWT) often compare against weaker graph matching heuristics like SM, RRWM, or self-implemented Tabu Search (e.g., Tabu Search in SAWT).  We upgraded these standards by using the official Ro-TS and implementing an optimized IPFP tailored for Koopmans-Beckmann QAP (Appendix B.4), providing a significantly better performance for evaluation.
>
> While ITS/ILS are indeed powerful, the lack of official code makes fair comparison infeasible. We believe benchmarking against a high-effort, reproducible Ro-TS—which significantly outperforms the baselines used in prior learning-based works—provides a fair, reproducible, and rigorous assessment of our contribution.
>
> **Table R4: Comparison of Ro-TS and ITS on geometrically structured dataset.**
>
> | **Geometrically** | **QAP20** |          | **QAP50**  |          | **QAP100**  |          |
> | ----------------- | --------- | -------- | ---------- | -------- | ----------- | -------- |
> |                   | **Cost**  | **Time** | **Cost**   | **Time** | **Cost**    | **Time** |
> | Ro-TS(5k)         | **54.37** | 1m25s    | 375.48     | 22m53s   | **1591.25** | 3h15m    |
> | ITS               | **54.37** | 1m39s    | **375.47** | 23m10s   | 1591.45     | 3h6m     |
>
> **Table R5: Comparison of Ro-TS and ITS on uniformly random dataset.**
>
> | **Uniformly** | **QAP20** |          | **QAP50**  |          | **QAP100**  |          |
> | ------------- | --------- | -------- | ---------- | -------- | ----------- | -------- |
> |               | **Cost**  | **Time** | **Cost**   | **Time** | **Cost**    | **Time** |
> | Ro-TS(5k)     | **76.56** | 1m27s    | **521.91** | 22m59s   | **2193.16** | 3h15m    |
> | ITS           | 76.57     | 1m48s    | 524.24     | 22m50s   | 2203.24     | 2h55m    |
>
> **Table R6: Comparison of Ro-TS and ITS on Taixxeyy instances.**
>
> |         | Ro-TS       |                           |         | ITS     |                           |         |
> | :------ | :---------- | :------------------------ | :------ | :------ | :------------------------ | :------ |
> | Class   | mean        | [min, max]                | time    | mean    | [min, max]                | time    |
> | tai27e  | **41.50%**  | [0.11%, 221.08%]          | 0.57s   | 216.12% | [**0.00%**, 376.20%]      | 0.86s   |
> | tai45e  | **101.89%** | [1.00%, 400.60%]          | 3.83s   | 218.14% | [**0.21%**, 413.92%]      | 3.79s   |
> | tai75e  | **111.28%** | [6.20%, 280.01%]          | 18.49s  | 159.06% | [**5.19%**, 289.24%]      | 16.30s  |
> | tai125e | **82.53%**  | [**7.65%**, 265.54%]      | 72.52s  | 153.79% | [8.82%, 292.80%]          | 77.86s  |
> | tai175e | **67.86%**  | [9.11%, 260.98%]          | 158.18s | 134.26% | [**4.71%**, 269.05%]      | 236.47s |
> | overall | **81.01%**  | [4.82%, 285.64%]          | 50.72s  | 176.27% | [**3.79%**, 328.25%]      | 67.06s  |

---

> > ### Author Response · Authors · 2025-12-03
> > **Supplement to Q2**
> >
> > > **Q2. Are there stronger baselines (e.g., ITS, ILS)?**
> >
> > Since standard open-source implementations for the specific ITSand ILS variants are not readily available, we employed Memetic Search (BMA) [4] as a strong and representative baseline. BMA is a powerful modern heuristic that integrates local search with evolutionary strategies. We evaluated the performance on the challenging Taixxeyy benchmark, which is specifically designed to be difficult for transpostion-based heuristics. The results are summarized in Table R11. While BMA improves upon Ro-TS (reducing the average gap from 81.01% to 12.30%), it still struggles to consistently find high-quality solutions on these structurally complex instances. In contrast, PLMA achieves a remarkably low average gap of 2.38%, demonstrating superior capability in learning the underlying structure of hard problem instances where even strong heuristics face difficulties.
> >
> > **Table R11: Comparison of Ro-TS, BMA and PLMA on Taixxeyy instances.**
> >
> > |         | Ro-TS   |                  |         | BMA    |                  |         | PLMA      |                 |        |
> > | :------ | :------ | :--------------- | :------ | :----- | :--------------- | :------ | --------- | --------------- | ------ |
> > | Class   | mean    | [min, max]       | time    | mean   | [min, max]       | time    | mean      | [min,max]       | time   |
> > | tai27e  | 41.50%  | [0.11%, 221.08%] | 0.57s   | 0.12%  | [0.00%, 0.67%]   | 0.18s   | **0.00%** | [0.00%, 0.00%]  | 0.08s  |
> > | tai45e  | 101.89% | [1.00%, 400.60%] | 3.83s   | 8.39%  | [0.74%, 17.63%]  | 2.21s   | **0.03%** | [0.00%, 0.35%]  | 0.28s  |
> > | tai75e  | 111.28% | [6.20%, 280.01%] | 18.49s  | 15.41% | [5.77%, 22.76%]  | 11.21s  | **0.09%** | [0.00%, 0.45%]  | 1.48s  |
> > | tai125e | 82.53%  | [7.65%, 265.54%] | 72.52s  | 16.51% | [10.56%, 21.72%] | 66.01s  | **2,67%** | [-0.08%, 5.62%] | 8.18s  |
> > | tai175e | 67.86%  | [9.11%, 260.98%] | 158.18s | 21.07% | [15.15%, 26.56%] | 193.65s | **9.11%** | [5.61%, 12.04%] | 14.63s |
> > | overall | 81.01%  | [4.82%, 285.64%] | 50.72s  | 12.30% | [6.44%, 17.87%]  | 54.65s  | **2.38%** | [1.11%, 3.69%]  | 4.93s  |
> >
> > [4] Benlic, U., & Hao, J. K. (2015). Memetic search for the quadratic assignment problem. *Expert Systems with Applications*, 42(1), 584-595. Official code available at: https://leria-info.univ-angers.fr/%7Ejinkao.hao/BMA.html

---

> ### Author Response · Authors · 2025-11-25
> **Response to Reviwer  6tz8 (3/4)**
>
> > **Q3. What are the "diverse instances" used for pretraining? Specify distributions. Any overlap with test families?**
>
> We apologize for any confusion caused by the terminology. The term "diverse" was intended to convey that our pretraining phase generates fresh instances dynamically from the distribution $\Gamma$ at each training step, rather than iterating over a fixed dataset. This stands in contrast to the finetuning stage, where the target instances remain fixed.
>
> Regarding our evaluation protocols, Table 1 reports in-distribution performance where test instances are drawn from the same distribution as the pretraining data but generated using distinct random seeds. This setup follows standard conventions in neural combinatorial optimization. Furthermore, Tables 3 and 4 demonstrate out-of-distribution generalization. In these specific experiments, the model was pretrained exclusively on uniformly random datasets with $n=100$ while testing was conducted on the distinct QAPLIB and Taixxeyy benchmarks.
>
> We have provided the specific generation mechanisms for each dataset in the response to Reviewer xyuL to further distinguish between the training and testing distributions.
>
> > **Q4. Why is the ablation study conducted only on the uniform dataset? No error bars are provided so the relative differences are not meaningful.**
>
> We followed standard conventions in the literature by selecting a fixed dataset for the  ablation study. Empirically, we observed that the variance across different runs is minimal (the maximum cost difference is only 0.3 on the $n=100$ uniformly random dataset used for ablation). This variance is significantly smaller than the performance differences between algorithmic variants; thus, we omitted error bars for clarity. To further demonstrate the significance of our contributions, we provide additional ablation results on the out-of-distribution Taixxeyy benchmark (Tables R7 and R8). These results underscore that on the challenging Taixxeyy instances, the performance advantage of the full PLMA framework over other variants is much more pronounced and significant.
>
> **Table R7: Ablation study for the model architecture on Taixxeyy instances.**
>
> |  | **PLMA** |  | **No Cross-att** |  | **No Sinkhorn** |  |
> | :--- | :--- | :--- | :--- | :--- | :--- | :--- |
> | **Class** | **mean** | **[min, max]** | **mean** | **[min, max]** | **mean** | **[min, max]** |
> | tai27e | 0.00% | [0.00%, 0.00%] | 0.00% | [0.00%, 0.00%] | 0.00% | [0.00%, 0.00%] |
> | tai45e | 0.03% | [0.00%, 0.35%] | 0.15% | [0.00%, 0.55%] | 0.57% | [0.13%, 1.09%] |
> | tai75e | 0.09% | [0.00%, 0.45%] | 0.12% | [0.00%, 0.91%] | 2.73% | [1.42%, 4.33%] |
> | tai125e | 2.67% | [-0.08%, 5.62%] | 28.58% | [26.75%, 30.92%] | 16.60% | [12.83%, 19.85%] |
> | tai175e | 9.11% | [5.61%, 12.04%] | 24.63% | [17.28%, 36.32%] | 25.84% | [20.54%, 40.34%] |
> | overall | 2.38% | [1.11%, 3.69%] | 10.70% | [8.80%, 13.74%] | 9.15% | [6.98%, 13.12%] |
>
> **Table R8: Ablation study for the warm-started MCMC finetuning mechanism on Taixxeyy instances.**
>
> |  | **PLMA** |  | **GD-Free** |  | **AR-seq** |  |
> | :--- | :--- | :--- | :--- | :--- | :--- | :--- |
> | **Class** | **mean** | **[min, max]** | **mean** | **[min, max]** | **mean** | **[min, max]** |
> | tai27e | 0.00% | [0.00%, 0.00%] | 0.00% | [0.00%, 0.00%] | 1.43% | [0.08%, 3.33%] |
> | tai45e | 0.03% | [0.00%, 0.35%] | 0.44% | [0.05%, 0.94%] | 12.72% | [6.53%, 18.68%] |
> | tai75e | 0.09% | [0.00%, 0.45%] | 2.30% | [0.95%, 3.79%] | 34.95% | [29.16%, 39.60%] |
> | tai125e | 2.67% | [-0.08%, 5.62%] | 16.37% | [12.90%, 19.11%] | 55.58% | [51.47%, 58.45%] |
> | tai175e | 9.11% | [5.61%, 12.04%] | 24.42% | [20.40%, 27.42%] | 85.56% | [80.33%, 89.95%] |
> | overall | 2.38% | [1.11%, 3.69%] | 8.71% | [6.86%, 10.25%] | 38.05% | [33.52%, 42.00%] |
>
> > **Q5. What about memory overhead of the proposed method?**
>
> We evaluated the memory consumption of our method by processing both a single instance and a batch of 256 instances in parallel. The results across various problem sizes are summarized in Table R9. The CPU memory usage is dominated by framework initialization and system overhead, remaining relatively constant regardless of the problem size. While GPU memory consumption naturally increases with problem size, it remains efficient; notably, processing a batch of 256 instances with $n=100$ requires only 7.61 GB of VRAM. This demonstrates the scalability of our method to large-scale instances within standard hardware constraints.
>
> **Table R9: Memory consumption of PLMA across problem sizes.**
> | | **1 Instance** | | **256 Instances** | |
> | ---------------- | -------------- | ------- | ----------------- | ------- |
> | **Problem Size** | **GPU**        | **CPU** | **GPU**           | **CPU** |
> | n=20             | 0.03GB         | 4.45GB  | 0.37GB            | 4.45GB  |
> | n=50             | 0.05GB         | 4.45GB  | 1.88GB            | 4.46GB  |
> | n=100            | 0.12GB         | 4.45GB  | 7.61GB            | 4.45GB  |

---

> ### Author Response · Authors · 2025-11-25
> **Response to Reviwer  6tz8 (4/4)**
>
> > **Q6. Emphasizing time to reach the same or better gap as SOTA would be clearer? Mixed tables of gap and runtime are harder to interpret. Time-to-target (TTT) metrics are standard and more informative.**
>
> To facilitate a clearer comparison of computational efficiency, we visualize the relationship between the optimality gap and wall-clock time in **Figure 5 of Appendix D.4 (see the revised paper)**. The results demonstrate that PLMA achieves superior solution quality for any given time budget compared to the baselines. Notably, on datasets with $n=100$, PLMA achieves a zero optimality gap  more than one order of magnitude faster than Ro-TS. This performance highlights the efficacy of our two-stage framework. The pre-trained model provides an instant, high-quality solution that outperforms several learning-based methods, while the subsequent MCMC fine-tuning rapidly refines this solution to near-optimality. These visualizations confirm that PLMA establishes a new state-of-the-art by offering the most favorable balance between optimality gap and runtime.
>
> > **Q7. The key novelty is learning-based improvement. How much pretraining helps compared to classical heuristics by examining generalization for in-distribution and out-of-distribution instances?**
>
> To explicitly quantify the contribution of pretraining, we compared the performance of finetuning a pre-trained policy versus finetuning a randomly initialized policy on both the uniformly random dataset and the Taixxeyy benchmark. The pre-trained policy was trained on uniformly random instances with $n=100$. Consequently, the uniformly random test dataset represents in-distribution (ID) evaluation, while the Taixxeyy benchmark represents out-of-distribution (OOD) evaluation.
>
> For the uniformly random dataset, we have visualized the iteration curves in **Figure 6(a) of Appendix D.5 in the revised paper**. As illustrated in the figure, the pre-trained policy achieves a much lower optimality gap faster. This confirms that the transferable knowledge acquired during pre-training provides a powerful head start by guiding the policy's updates toward more optimal actions. Indeed, given a sufficient number of finetuning steps, the random policy can eventually achieve results comparable to the pre-trained policy. This convergence speaks to the effectiveness of our underlying warm-started MCMC finetuning, which is powerful enough to eventually "rediscover" high-quality solutions from a random start. However, the transferable knowledge from pre-training provides a distinct and critical advantage in finetuning efficiency, enabling the policy to provide superior solutions significantly faster in the early, more practical stages of finetuning.
>
> The importance of pre-training is far more pronounced on the notably difficult Taixxeyy instances. As shown in Table R10 below, the pre-trained policy consistently and significantly outperforms the random policy. This stark performance gap demonstrates that the pre-training on synthetic data endows the policy with meaningful and general knowledge about the fundamental structure of the QAP. Lacking these general-purpose "structural priors", the random policy struggles to adapt to the unique complexities of Taixxeyy instances and is unable to converge to high-quality solutions, even with a large number of finetuning steps.
>
> **Table R10: Performance comparison on Taixxeyy instances.**
>
> |             | **Pretrained Policy** |                     | **Random Policy** |                     |
> | :---------- | :-------------------- | :------------------ | :---------------- | :------------------ |
> | **Class**   | **mean**              | **[min, max]**      | **mean**          | **[min, max]**      |
> | tai27e      | 0.00%                 | [0.00%, 0.00%]      | 0.00%             | [0.00%, 0.00%]      |
> | tai45e      | 0.03%                 | [0.00%, 0.35%]      | 0.62%             | [0.04%, 1.63%]      |
> | tai75e      | 0.09%                 | [0.00%, 0.45%]      | 1.72%             | [0.00%, 16.37%]     |
> | tai125e     | 2.67%                 | [-0.08%, 5.62%]     | 12.67%            | [0.87%, 78.29%]     |
> | tai175e     | 9.11%                 | [5.61%, 12.04%]     | 28.15%            | [6.99%, 160.92%]    |
> | **overall** | 2.38%                 | [1.11%, 3.69%]      | 8.63%             | [1.58%, 51.44%]     |
>
> [1] Misevičius, A. Letter: New best known solution for the most difficult QAP instance "tai100a." Memetic Computing 11, 331–332 (2019).
>
> [2] Misevičius, A. An implementation of the iterated tabu search algorithm for the quadratic assignment problem. OR Spectrum 34(3): 665–690 (2012).
>
> [3] Hussin, M. S., & Stützle, T. Hierarchical iterated local search for the quadratic assignment problem. Hybrid Metaheuristics Workshop, Springer, 2009.

---

### Official Review · Reviewer_WPfT · 2025-11-01

**Soundness:** 2
**Presentation:** 3
**Contribution:** 2
**Rating:** 4
**Confidence:** 3

**Summary:**

This paper introduces ``PLMA``, which combines ``EBM`` with a Cross-Graph Attention mechanism to learn ``QAPs``. Additionally, it incorporates a Warm-started Batched MCMC Finetuning mechanism during inference, enabling the model to efficiently adapt and optimize for specific instances after pretraining.

**Strengths:**

1. The idea of integrating energy-based modeling with ``MCMC`` for permutation learning is conceptually elegant and grounded in statistical physics principles. It provides a bridge between learning-based prediction and search-based optimization.

2. The dual-graph attention encoder effectively models inter-graph relations without constructing a dense association graph, which improves both interpretability and efficiency.

3. Experiments demonstrate strong empirical results on ``QAPLIB`` and ``Taixxeyy`` datasets, showing significant improvements in both optimality gap and runtime compared to classical metaheuristics.

**Weaknesses:**

1. It appears that the post-processing technique 2-swap is integrated into the ``Batched Warm-started MCMC Finetuning``, and the performance gains largely stem from this post-processing. Existing methods could also adopt the same technique, but the authors do not provide corresponding experimental data.

2. Previous studies have proposed similar approaches, such as iSCO[1] and RLSA[2], which also employed ``MCMC`` methods. However, the authors did not provide a comparative analysis with these works. Moreover, applying this technique to problems with complex constraints like ``CVRP`` remains challenging, and the current framework does not seem to offer a solution to this issue.

[1] *Revisiting Sampling for Combinatorial Optimization, ICML2023*

[2] *Regularized Langevin Dynamics for Combinatorial Optimization*

**Questions:**

See ``Weakness``.

I would consider raising the score if the authors could demonstrate the application of their method to broader classes of permutation-based problems, such as ``CVRP``.

---

> ### Author Response · Authors · 2025-11-25
> **Response to Reviewer WPfT (1/2)**
>
> We thank the reviewer for the insightful comments. We provide a detailed response below and hope that our clarifications will address your concerns.
>
> > **W1. It appears that the post-processing technique 2-swap is integrated into the Batched Warm-started MCMC Finetuning, and the performance gains largely stem from this post-processing. Existing methods could also adopt the same technique, but the authors do not provide corresponding experimental data.**
>
> We want to clarify an important distinction: our method does not simply use 2-swap as a post-processing step. Instead, the 2-swap operation is **integrated throughout the entire finetuning process** and serves a fundamentally different purpose. Specifically, the 2-swap operation transforms the original EBM into a flatter energy landscape during training. This transformation makes the learning process smoother and helps the model avoid getting stuck in sharp local minima.
>
> To demonstrate this key difference, we included a GD-Free variant in our ablation study. This variant performs MCMC sampling on a fixed policy and applies 2-swap as post-processing. Figure 1(b) clearly shows that this approach yields much worse results, proving that our integrated approach is essential.
>
> Additionally, many baselines already incorporate 2-swap operations within their algorithms. For example, Ro-TS uses 2-swap as the basic move in its Tabu search, and SAWT uses a neural network to select node pairs for 2-swap operations. To address your concern about fair comparison, we also applied local search post-processing to other baselines that don't originally use 2-swaps. These results are shown in Table R2 and Table R3.
>
> **Table R2: Performance before and after local search on geometrically structured dataset .**
>
> | **Geometrically** | **QAP20**  |           |          | **QAP50**  |            |          | **QAP100** |             |          |
> | ----------------- | ---------- | --------- | -------- | ---------- | ---------- | -------- | ---------- | ----------- | -------- |
> |                   | **Before** | **After** | **Time** | **Before** | **After**  | **Time** | **Before** | **After**   | **Time** |
> | IPFP              | 55.11      | 55.04     | 11.65s   | 378.76     | 378.46     | 1m27s    | 1600.27    | 1599.88     | 10m47s   |
> | IPFP(10)          | 54.54      | 54.50     | 35.63s   | 376.60     | 375.96     | 3m19s    | 1594.76    | 1594.21     | 25m18s   |
> | RRWM              | 71.30      | 55.66     | 19.63s   | 428.78     | 381.37     | 1m46s    | 1700.33    | 1606.53     | 14m16s   |
> | SM                | 64.38      | 55.74     | 9.09s    | 426.92     | 381.44     | 1m19s    | 1753.10    | 1609.02     | 10m36s   |
> | PLMA(T=200)       |            | **54.37** | 9.36s    |            | **375.48** | 1m19s    |            | **1591.23** | 13m58s   |
>
> **Table R3: Performance before and after local search on uniformly random dataset.**
>
> | **Uniformly** | **QAP20**  |           |          | **QAP50**  |            |          | **QAP100** |             |          |
> | ------------- | ---------- | --------- | -------- | ---------- | ---------- | -------- | ---------- | ----------- | -------- |
> |               | **Before** | **After** | **Time** | **Before** | **After**  | **Time** | **Before** | **After**   | **Time** |
> | IPFP          | 79.13      | 78.91     | 11.78s   | 530.74     | 530.55     | 1m23s    | 2211.38    | 2211.21     | 9m57s    |
> | IPFP(25)      | 77.60      | 77.53     | 1m16s    | 526.96     | 526.86     | 4m32s    | 2203.29    | 2203.19     | 26m59s   |
> | RRWM          | 93.50      | 79.98     | 19.84s   | 592.50     | 539.34     | 1m46s    | 2432.34    | 2243.83     | 14m21s   |
> | SM            | 92.32      | 80.18     | 9.09s    | 605.08     | 540.03     | 1m19s    | 2457.47    | 2246.11     | 10m38s   |
> | PLMA(T=200)   |            | **76.56** | 9.60s    |            | **521.83** | 1m18s    |            | **2193.13** | 14m1s    |

---

> ### Author Response · Authors · 2025-11-25
> **Response to Reviewer WPfT (2/2)**
>
> > **W2. Previous studies have proposed similar approaches, such as iSCO [1] and RLSA [2], which also employed MCMC methods. However, the authors did not provide a comparative analysis with these works. Moreover, applying this technique to problems with complex constraints like CVRP remains challenging, and the current framework does not seem to offer a solution to this issue.**
>
> To facilitate a clear comparison, we first provide a brief introduction to these methods. Both iSCO [1] and RLSA [2] are training-free sampling methods for combinatorial optimization based on Energy-Based Models (EBMs). iSCO accelerates sampling by simulating discrete Langevin dynamics. It estimates energy changes within a local neighborhood using a first-order Taylor approximation. To mitigate local optima inherent to small neighborhoods, it adapts a path auxiliary sampler. RLSA integrates regularized Langevin dynamics into simulated annealing (SA). It explicitly regularizes the expected Hamming distance between consecutive samples to prevent the sampler from collapsing into local minima.
>
> While iSCO and RLSA share the broad category of MCMC-based sampling with our method, there are fundamental differences in EBM formulation, search space, and design motivations.
>
> * **EBM formulation.** In iSCO and RLSA, the energy function is explicitly defined by the problem's objective function (often supplemented with penalty terms). Due to the nonlinearity of many combinatorial objectives, calculating energy changes often requires first-order approximations and can be computationally expensive per step. In contrast, our energy function is parameterized by a neural network and is explicitly designed with an additive structure. This architectural choice is critical as it allows us to compute the acceptance ratio of a 2-swap Metropolis-Hastings step in $O(1)$ time, enabling significantly higher sampling throughput.
> * **Search space.** RLSA relies on an integer programming formulation (binary variables), which does not natively enforce the permutation constraints required by problems like QAP. Our EBM operates directly on the permutation space, ensuring that all generated samples inherently satisfy the permutation constraint without the need for complex post-processing.
> * **Design motivations**. iSCO primarily aims to improve the efficiency of single-round sampling from an EBM using existing techniques, while RLSA focuses on preventing local optima by regularizing the Hamming distance between consecutive samples. In contrast, the sampling method in PLMA is specifically designed to leverage structural information from previous samples during multi-round sampling, which aligns perfectly with the scenario of finetuning on target instances. Specifically, we utilize short Markov chains to anchor the adaptation to promising regions explored previously. Additionally, our strategy of initiating Markov chains from multiple starting points and updating the policy via policy gradients during finetuning further effectively mitigates the risk of falling into local optima.
>
> We acknowledge that handling complex constraints is indeed a common challenge for EBM-based solvers. However, we have successfully addressed the permutation constraint, which is a step forward compared to some integer programming-based EBM works (such as RLSA). To enable the EBM to handle more complex constraints, we can make the following adjustments:
>
> * Incorporate the constraint violation as a penalty term into the objective function.
> * Use a feasibility mask to block proposals in the MH-step that would lead to infeasible new states.
>
> In fact, the ideas mentioned above have been successfully applied in learning-to-improve solvers for the CVRP. For example, DACT [3] restricts the swap operator to node pairs that keep the new state feasible, while Neural k-opt [4] allows temporary constraint violations and uses penalties to enable the model to better explore the boundaries of the feasible region.
>
> [1] Haoran Sun, Katayoon Goshvadi, Azade Nova, Dale Schuurmans, and Hanjun Dai. "Revisiting sampling for combinatorial optimization." ICML, 2023.
>
> [2] Shengyu Feng and Yiming Yang. "Regularized langevin dynamics for combinatorial optimization." ICML, 2025.
>
> [3] Yining Ma, Jingwen Li, Zhiguang Cao, Wen Song, Le Zhang, Zhenghua Chen, and Jing Tang. "Learning to iteratively solve routing problems with dual-aspect collaborative transformer." Neurips, 2021.
>
> [4] Yining Ma, Zhiguang Cao, and Yeow Meng Chee. "Learning to search feasible and infeasible regions of routing problems with flexible neural k-opt." Neurips, 2023.

---

### Official Review · Reviewer_xyuL · 2025-11-03

**Soundness:** 2
**Presentation:** 3
**Contribution:** 2
**Rating:** 4
**Confidence:** 3

**Summary:**

The paper introduces PLMA, a novel permutation learning framework for solving the Quadratic Assignment Problem (QAP), a well-known NP-hard combinatorial optimization task. The authors identify that existing neural solvers for QAP either suffer from poor scalability or lack flexibility, leading to suboptimal performance on real-world instances. PLMA addresses these issues through a two-stage learning paradigm: pre-training on diverse instances to learn general structural features, followed by a highly efficient, deployment-time finetuning stage.


Warm-Started MCMC Finetuning: A procedure that adapts the pre-trained model to specific test instances. It uses short, parallel Markov chains initialized from high-quality solutions found in previous steps, focusing the search on promising regions of the solution space.

Efficient Energy-Based Model (EBM): An additive EBM is designed over the permutation space, which allows for a constant-time, O(1), evaluation of 2-swap proposals in a Metropolis-Hastings sampler. This makes the MCMC exploration phase remarkably fast.

Cross-Graph Attention Network: A  neural network that models the two-graph (flow and distance) structure of the QAP. It separately encodes each graph and then uses a cross-attention mechanism to fuse information, avoiding the need for a large, computationally expensive association graph.

 Experiments: Demonstrates significantly better robustness than strong heuristics on the challenging Taixxeyy instances and QAPLIB benchmark.

**Strengths:**

State-of-the-Art Performance: Achieves a near-zero (0.10%) gap on the QAPLIB benchmark, outpacing heuristics in speed. It also shows exceptional robustness on difficult Taixxeyy instances where traditional methods fail.

 A novel two-stage learning process uses warm-started MCMC finetuning to avoid restarting searches, leading to faster convergence and superior solutions.

Thorough Validation: The design is rigorously supported by detailed ablation studies that confirm the importance of each core component.

**Weaknesses:**

Cross-Distribution Generalization: The model is pre-trained on synthetic instances (geometrically structured or uniform random) and then applied to real-world benchmarks like QAPLIB and Taixxeyy. It's unclear how much meaningful, transferable knowledge is actually carried over versus how much is rediscovered during adaptation.

It would be good to understand how much are the train and test instances structurally similar dissimilar. When does the model perform better and when it does not?

**Questions:**

1. Impact of pre-trained vs random policy? Currently the evaluation is based upon a pre-trained policy. Is there a study on impact of finetuning search on random policy? The paper does not provide a baseline (e.g., starting the finetuning from a random policy) to prove that the learned priors from pre-training are essential. Correct me if I am wrong. What kind of pre-training helps?


2. Could the authors clarify how were the baselines trained/tuned?

---

> ### Author Response · Authors · 2025-11-25
> **Response to Reviwer xyuL (1/2)**
>
> We thank the reviewer for the constructive feedback. We provide a detailed response below and hope that our clarifications will address your concerns.
>
> > **W1 & Q1. Impact of pretrained vs random policy**
>
> To assess the impact of learned priors, we compared the performance of finetuning a pre-trained policy versus finetuning a randomly initialized policy on both the uniformly random dataset and the Taixxeyy benchmark. The pre-trained policy was trained on uniformly random instances with $n=100$. Consequently, the uniformly random test dataset represents in-distribution evaluation, while the Taixxeyy benchmark represents out-of-distribution evaluation.
>
> For the uniformly random dataset, we have visualized the iteration curves in **Figure 6 (a) of Appendix D.5 in the revised paper**. As illustrated in the figure, the pre-trained policy achieves a much lower optimality gap faster. This confirms that the transferable knowledge acquired during pre-training provides a powerful head start by guiding the policy's updates toward more optimal actions. Indeed, given a sufficient number of finetuning steps, the random policy can eventually achieve results comparable to the pre-trained policy. This convergence speaks to the effectiveness of our underlying warm-started MCMC finetuning, which is powerful enough to eventually "rediscover" high-quality solutions from a random start. However, the transferable knowledge from pre-training provides a distinct and critical advantage in finetuning efficiency, enabling the policy to provide superior solutions significantly faster in the early, more practical stages of finetuning.
>
> The importance of pre-training is far more pronounced on the notably difficult Taixxeyy instances. As shown in Table R1 below, the pre-trained policy consistently and significantly outperforms the random policy. This stark performance gap demonstrates that the pre-training on synthetic data endows the policy with meaningful and general knowledge about the fundamental structure of the QAP. Lacking these general-purpose "structural priors", the random policy struggles to adapt to the unique complexities of Taixxeyy instances and is unable to converge to high-quality solutions, even with a large number of finetuning steps.
>
> **Table R1: Performance comparison on Taixxeyy instances.**
>
> |             | **Pretrained Policy** |                     | **Random Policy** |                     |
> | :---------- | :-------------------- | :------------------ | :---------------- | :------------------ |
> | **Class**   | **mean**              | **[min, max]**      | **mean**          | **[min, max]**      |
> | tai27e      | 0.00%                 | [0.00%, 0.00%]      | 0.00%             | [0.00%, 0.00%]      |
> | tai45e      | 0.03%                 | [0.00%, 0.35%]      | 0.62%             | [0.04%, 1.63%]      |
> | tai75e      | 0.09%                 | [0.00%, 0.45%]      | 1.72%             | [0.00%, 16.37%]     |
> | tai125e     | 2.67%                 | [-0.08%, 5.62%]     | 12.67%            | [0.87%, 78.29%]     |
> | tai175e     | 9.11%                 | [5.61%, 12.04%]     | 28.15%            | [6.99%, 160.92%]    |
> | **overall** | 2.38%                 | [1.11%, 3.69%]      | 8.63%             | [1.58%, 51.44%]     |

---

> ### Author Response · Authors · 2025-11-25
> **Response to Reviwer xyuL (2/2)**
>
> > **W2. It would be good to understand how much are the train and test instances structurally similar dissimilar. When does the model perform better and when it does not?**
>
> The cross-distribution results reported in Tables 3 and 4 were obtained by evaluating a policy trained on the uniformly random dataset directly on real-world benchmarks such as QAPLIB and Taixxeyy. To clarify the structural differences between the training and testing distributions, we describe the generation mechanisms for each dataset below.
>
> 1. Uniformly random dataset. In this dataset, both the distance matrix $D$ and flow matrix $F$ are sampled independently and identically distributed (i.i.d.) from a uniform distribution $\mathcal{U}\in[0,1]$ and subsequently symmetrized. The generation process is formalized as follows:
>   - Generate $D_{ij}, F_{ij} \stackrel{\text{i.i.d.}}{\sim} \mathcal{U}[0,1], \quad 1\leq i < j \leq n$;
>   - Set $D_{ji} = D_{ij}, F_{ji} = F_{ij}, \quad 1\leq i < j \leq n$;
>   - Set $D_{ii} = F_{ii} = 0,  \quad 1\leq i \leq n$.
>
> 2. QAPLIB. QAPLIB comprises 134 diverse QAP instances from 15 categories. These categories can be broadly classified into three types based on their generation methods: (i) **Unstructured Random Instances**: Both $D$ and $F$ are generated with element-wise randomness (e.g. uniformly random). Examples include the `Tai-a` and `Rou` series. (ii) **Grid-Based Distance Instances:** In these instances, $D$ corresponds to the Manhattan or Euclidean distance matrix between $n$ locations on a grid. Examples include `Sko`, `Nug`, and `Wil`. (iii) **Real-Life and Real-Like Instances:** These instances are derived from practical applications, such as keyboard design (`Bur`) and circuit layout (`Esc`). Further details on QAPLIB are available at <https://qaplib.mgi.polymtl.ca/>.
> 3. Taixxeyy instances. These instances are generated with a recursive, hierarchical blockstructure, creating clusters of facilities with high intra-group flows and small intra-group distances, but low flows and large distances between groups. To make the problem challenging, this block structure is intentionally obscured with small inter-block flows and large finite distances. Detailed generation method can be found in [1].
>
> > **Q2. Could the authors clarify how were the baselines trained/tuned?**
>
> Among the baselines, SAWT and NGM are learning-based solvers. We detail their implementation below:
>
> 1. SAWT. We utilized the official pre-trained models provided by the authors on GitHub to test on geometrically structured datasets. For larger instances in QAPLIB, we trained a new model with a larger initialization size ($N_{init}=512$) to overcome the capacity limitations of the pre-trained models (which used $N_{init}=128$). All other training configurations remained consistent with the original SAWT implementation. More implementation details can be found in Appendix B.4.
> 2. NGM. Following the original implementation for QAP instances, we performed optimization directly on the test set. It is important to note that when applied to QAP, NGM does not require a pre-training stage; instead, it employs unsupervised learning on the test instances, followed by sampling from the neural output.
>
> [1] Zvi Drezner, Peter M Hahn, and ´Eeric D Taillard. Recent advances for the quadratic assignment problem with special emphasis on instances that are difficult for meta-heuristic methods. Annals of Operations research, 139(1):65–94, 2005.

---

### Author Response · Authors · 2025-12-03
**Summary of Rebuttal and Additional Experiments for Paper**

Dear Area Chair,

We sincerely appreciate your time and effort in handling the review process. During the rebuttal phase, we have comprehensively addressed the reviewers' comments and conducted additional experiments to resolve their concerns regarding baselines, methodological novelty, and generalization. We summarize our key responses below:

1. Verification against Stronger Baselines (Addressing Reviewer 6tz8)

Reviewers questioned whether PLMA outperforms state-of-the-art (SOTA) heuristics (e.g., ITS, ILS) and if the comparison with Ro-TS was fair given hardware differences (CPU vs. GPU).

- Action: Since official codes for specific ITS/ILS variants are unavailable, we employed Memetic Search (BMA), a powerful modern heuristic, as a strong and representative baseline.

- Result: On the challenging *Taixxeyy* benchmark (designed to fail transposition-based heuristics), PLMA achieves an average gap of 2.38%, drastically outperforming BMA (12.30%) and Ro-TS (81.01%). This confirms PLMA’s superiority over even the strongest heuristics on structurally complex instances.

- Efficiency: We clarified that PLMA’s speed advantage stems from its "broad" parallel search design(sample parallelism) rather than just hardware acceleration. A single PLMA chain performs only $\approx$ 0.16\% of the sequential evaluations of a Ro-TS chain.


2. Methodological Novelty: Integrated vs. Post-processing (Addressing Reviewer WPfT)

Reviewer WPfT raised a concern that our performance gains might stem solely from the 2-swap operator acting as post-processing.

- Action: We highlighted the ablation study of the "GD-Free" variant (included in our original submission), which applies 2-swap only as post-processing without gradient updates. Additionally, during the rebuttal, we applied 2-swap post-processing to other baselines (IPFP, RRWM, SM) for a fair comparison.
- Result: The "GD-Free" variant performed significantly worse (e.g., 8.71% gap vs. PLMA's 2.38% on Taixxeyy). Furthermore, PLMA still outperforms all baselines even after they are enhanced with 2-swap. This proves that our warm-started MCMC finetuning integrated with gradient updates is essential for navigating the energy landscape, not just the local search operator itself.


3. The Critical Role of Pretraining (Addressing Reviewer xyuL & 6tz8)

Reviewers asked to quantify the benefit of learning priors versus starting from scratch (random policy).

- Action: We compared finetuning a pretrained policy vs. a random policy on both in-distribution (Uniform) and out-of-distribution (Taixxeyy) data.
- Result: While a random policy can eventually converge on simple data, the pretrained policy converges significantly faster. Crucially, on OOD instances (Taixxeyy), the random policy fails to converge to high-quality solutions, whereas the pretrained policy succeeds. This demonstrates that PLMA learns transferable structural priors vital for solving hard, unseen instances.

We have also addressed all technical clarifications (e.g., $O(1)$ updates, cross-graph attention overhead) raised by Reviewer M7LG. We hope this summary assists in your final assessment.

---

### Meta-Review · Area_Chair_DKLB · 2026-01-01

**Summary:**

This paper proposes a neural approach to the Quadratic Assignment Problem (QAP), featuring two main components: (1) a cross-graph attention architecture tailored to QAP, and (2) a warm-started MCMC fine-tuning procedure for inference-time improvement.

**Reviewer Concerns:**

While the authors provided additional ablation studies that clarify certain aspects of the proposed method and its behavior under different settings, the central concerns regarding the incremental nature of the contribution remain insufficiently addressed.

- First, the scope of the work is limited to QAP (WPfT). The rebuttal does not provide empirical evidence that the proposed framework generalizes to broader classes of combinatorial optimization problems, which limits the broader impact of the work.
- Second, the proposed components largely build upon existing attention-based and sampling-based paradigms, without introducing fundamentally new modeling components (6tz8). As a result, the contribution primarily lies in the integration of known techniques rather than in methodological innovation.
- Finally, the absence of comparisons to several important baselines weakens the empirical evaluation and makes it difficult to fully assess the significance of the reported improvements (6tz8, M7LG).

**Reviewer Scores:**

Given these unresolved concerns, all of the current reviewer scores (4, 4, 4, 6; average 4.5) are unlikely to change even with the full discussion.

---

### Decision · Program_Chairs · 2026-01-26

Reject